# Simultaneous stabilization of actin cytoskeleton in multiple nephron-specific cells protects the kidney from diverse injury

Kamalika Mukherjee [1,7], Changkyu Gu[1,7], Agnieszka Collins [1,7], Marcel Mettlen[2], Beata Samelko[3], Mehmet M. Altintas [3], Yashwanth R. Sudhini[3], Xuexiang Wang [3], Richard Bouley [1], Dennis Brown[1], Bradley P. Pedro[1], Susan L. Bane[4], Vineet Gupta [3], Paul T. Brinkkoetter[5,6], Henning Hagmann[5,6], Jochen Reiser[3✉] & Sanja Sever [1✉]

Chronic kidney diseases and acute kidney injury are mechanistically distinct kidney diseases. While chronic kidney diseases are associated with podocyte injury, acute kidney injury affects renal tubular epithelial cells. Despite these differences, a cardinal feature of both acute and chronic kidney diseases is dysregulated actin cytoskeleton. We have shown that pharmacological activation of GTPase dynamin ameliorates podocyte injury in murine models of chronic kidney diseases by promoting actin polymerization. Here we establish dynamin's role in modulating stiffness and polarity of renal tubular epithelial cells by crosslinking actin filaments into branched networks. Activation of dynamin's crosslinking capability by a small molecule agonist stabilizes the actomyosin cortex of the apical membrane against injury, which in turn preserves renal function in various murine models of acute kidney injury. Notably, a dynamin agonist simultaneously attenuates podocyte and tubular injury in the genetic murine model of Alport syndrome. Our study provides evidence for the feasibility and highlights the benefits of novel holistic nephron-protective therapies.

[1] Department of Medicine, Harvard Medical School and Division of Nephrology, Massachusetts General Hospital, Boston, MA, USA. [2] Department of Cell Biology, University of Texas Southwestern Medical Center, Dallas, TX, USA. [3] Department of Medicine, Rush University Medical Center, Chicago, IL, USA. [4] Department of Chemistry, Binghamton University, State University of New York, Binghamton, NY, USA. [5] Department of Internal Medicine-Center for Molecular Medicine Cologne, University of Cologne and Faculty of Medicine-University Hospital Cologne, Cologne, Germany. [6] Cologne Cluster of Excellence on Cellular Stress Responses in Ageing-Associated Diseases (CECAD) and Systems Biology of Ageing Cologne (Sybacol), Cologne, Germany. [7] These authors contributed equally: Kamalika Mukherjee, Changkyu Gu, Agnieszka Collins. ✉email: Jochen_reiser@rush.edu; ssever@mgh.harvard.edu

The leading causes of acute kidney injury (AKI) are ischemia, hypoxia, or nephrotoxicity[1]. While it can be reversed, AKI represents a significant healthcare problem with high mortality and no definitive treatment. Regardless of its etiology, AKI primarily injures polarized epithelial cells of the renal tubules whose apical microvilli form the tubular brush border that participates in coordinating essential electrolyte and water transport[2]. An early morphological feature of AKI is loss of the brush border and cell polarity due to the breakdown of the actomyosin cortex at the apical membrane[1].

The establishment and maintenance of cell polarity involve signaling cascades, membrane trafficking, and cytoskeletal dynamics, all of which are highly coordinated[3]. The organization of the apical membrane is largely determined by the architecture of the actomyosin networks[4], which establishes cortex stiffness, thus facilitating the clustering of polarity proteins. While myosin II motors are considered the primary generator of cortical stiffness[5,6], the architecture of the actomyosin cortex is established by a myriad of actin-binding proteins (ABPs)[7].

In addition to known ABPs, the brush border of renal tubules is highly enriched in dynamin[8], a GTPase best known for its role in endocytosis[9]. Dynamin has an intrinsic propensity to assemble into multiple oligomerization states such as dimers, tetramers, rings, and spirals[9]. We identified for the first time direct dynamin–actin interactions[10] and showed that dynamin's oligomerization regulates actin polymerization in podocytes[11,12], specialized cells essential for the selectivity of the kidney filter. Using murine models of CKD, we have shown that activation of dynamin-dependent actin polymerization reverses podocyte injury by restoring their unique structure and function[12].

Here we show that in renal tubular epithelial cells dynamin cross-links filamentous actin (F-actin) into branched networks. Dynamin's cross-linking capability is defined by its oligomerization state and the length of F-actin. Pharmacological activation of dynamin oligomerization counteracts AKI by stabilizing the actin networks and thus cell integrity, which partially protects renal epithelial cells from oxidative stress-induced injury. Our study identifies the actomyosin cortex of the apical membrane of the renal tubular cell as a druggable target in AKI via dynamin as a proxy.

## Results

**Dynamin oligomerization establishes the stiffness and morphology of the apical membrane.** To examine the role of dynamin–actin interactions in polarized renal tubular epithelial cells, we utilized Bis-T-23, an allosteric activator of actin-dependent dynamin oligomerization in a reconstituted system[13], in the cells[11,13], and in the whole organism[12]. Cellular phenotypes were assessed in Madin-Darby Canine Kidney (MDCK) cells by following the status of the F-actin and the staining pattern of a tight junction protein zonula occludens-1 (ZO-1), which is considered a biomarker of cell polarity. Cytochalasin D (CytoD) and latrunculin A (LatA), known inhibitors of actin polymerization, decreased F-actin levels, and induced discontinuous ZO-1 staining (Supplementary Fig. 1a). In contrast, Bis-T-23 induced a slight increase in F-actin levels without any effect on ZO-1 staining. Addition of Bis-T-23 prior to but not after LatA, partially preserved F-actin levels and cell polarity. Neither DMSO vehicle nor dynamin inhibitor dynole[14] exhibited any effect (Supplementary Fig. 1a).

Scanning electron microscopy (SEM) allowed us to visualize drug-induced alterations of cell morphology focusing on the apical membrane (Fig. 1a). The average MDCK cell height was $11 \pm 2$ μm, and the average length of the microvilli was $0.63 \pm 0.2$ μm (Table 1), which is within the range observed in

the kidney[15]. LatA decreased cell height, microvilli length and shifted the uniformly distributed microvilli into clusters, whereas Bis-T-23 induced the opposite effects (Table 1, Fig. 1a). When added prior to LatA, Bis-T-23 partially preserved cell height and microvilli length. Since microvilli exhibit exquisite length control defined by the cortical actin at their base[16], these data provide evidence that Bis-T-23 modified the actomyosin cortex at the apical membrane.

To determine the exact effect that Bis-T-23 had on the cortical actin, we visualized the actomyosin cortex within the lamellipodia using platinum replica electron microscopy (PR-EM). LatA decreased the density of the actin networks, and this effect was partially abrogated by the addition of Bis-T-23 prior to LatA (Fig. 1b). As LatA accelerates actin filament depolymerization by sequestering actin monomers, we next examined whether the observed preservation of actomyosin cortex by Bis-T-23 was due to its positive effect on actin polymerization. In contrast to the potent stimulation of actin polymerization observed in podocyte cell extracts[10,11], Bis-T-23 only marginally increased actin polymerization in MDCK cell extract (Supplementary Fig. 1b). Similarly, immunodepletion of endogenous dynamin-2 (Dyn2) from the extract or inhibition of its GTPase activity by dynole resulted in marginal impairment of actin polymerization (Supplementary Fig. 1c). While LatA and CytoD significantly impaired actin polymerization, the addition of Bis-T-23 prior to LatA or CytoD was not able to overcome their inhibitory effects (Supplementary Fig. 1d, e). Jasplakinolide, a drug that induces actin polymerization by stimulating actin filament nucleation[17], did not significantly increase the overall level of polymerization (Supplementary Fig. 1d), suggesting that MDCK cell lysate exhibit a near maximal level of polymerized actin. Together, these data indicated that the effects of Bis-T-23 on the morphology of the apical membrane in MDCK cells were driven by a mechanism other than actin polymerization.

Given the common knowledge of enriched localization of Dyn2 and F-actin at the brush border of renal epithelial cells[8], and dynamin's role in endocytosis, we next investigated if Bis-T-23 was affecting actin indirectly via alterations in endocytosis. As expected, both Dyn2 and F-actin co-localized at the actomyosin cortex underneath the apical membrane, within the microvilli, and at clathrin-coated pits (CCPs), defined by their distinct shape and size (Supplementary Fig. 1f). We examined the dynamics of CCPs using total internal reflection fluorescence (TIRF) microscopy[18,19]. Bis-T-23, even at its highest concentration, had no effect on the distribution of the CCPs' lifetimes, whereas dynole decreased the number of productive CCPs (Supplementary Fig. 1g). This lack of correlation between the level of endocytosis and alterations in cell morphology asserts that Bis-T-23 targets the cortical actin without influencing dynamin's role in endocytosis.

Since renal cell polarity is maintained by the architecture and sustained contraction of the actomyosin networks, which establishes cell stiffness at the apical membrane[20], we next measured cell stiffness using atomic force microscopy (AFM). Nanowizard IV system and JPK analysis software were used to determine changes in Young's Modulus[21] under different experimental settings (Supplementary Fig. 2a). Treatment with LatA resulted in a significant decrease in cell–cell contact stiffness and apical cell stiffness in MDCK cells (Fig. 1c–e). In contrast, Bis-T-23 significantly increased cell stiffness compared to the DMSO vehicle (Fig. 1c–e), consistent with its positive effects on cell height, microvilli number, and the density of actin networks (Table 1 and Fig. 1b)[22]. The addition of Bis-T-23 prior to LatA strongly reduced the negative effect of LatA on cell stiffness (Fig. 1c–e), in accordance with Bis-T-23's positive effect on actin networks and apical cell morphology (Table 1, Fig. 1b).

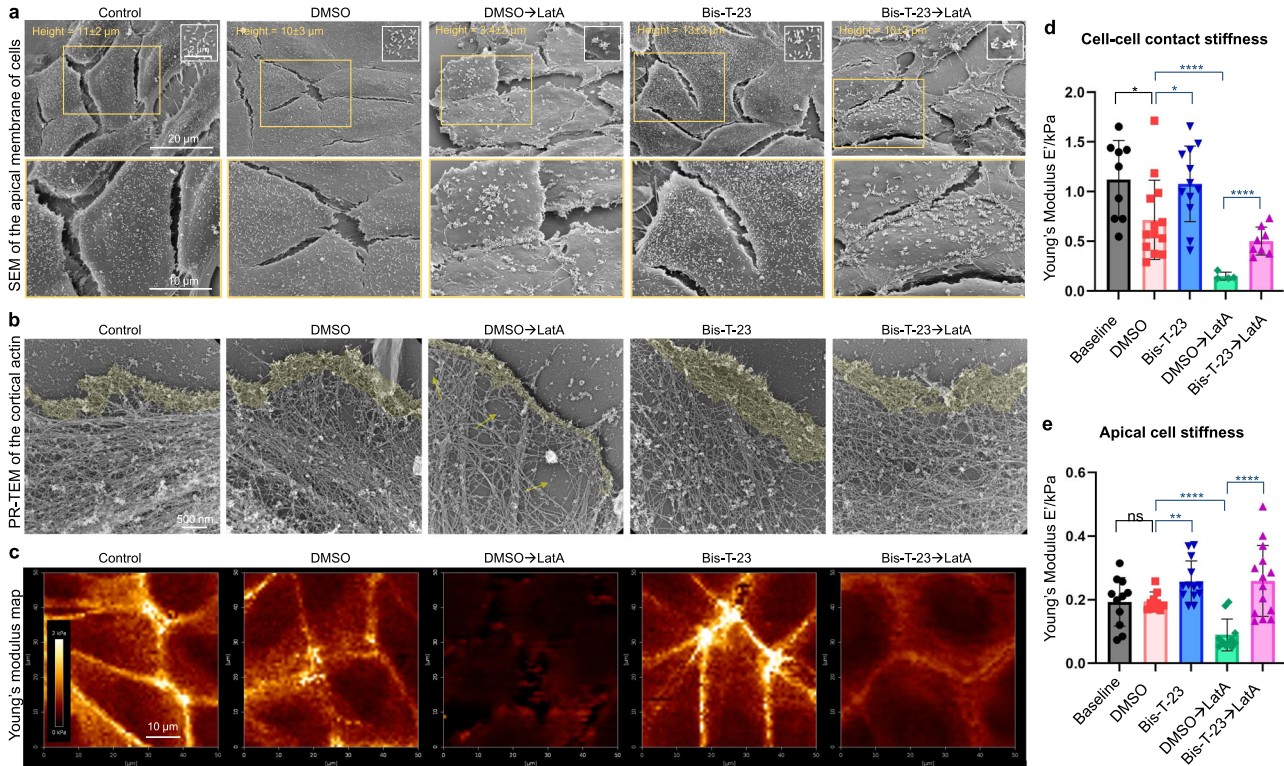

**Fig. 1 Dynamin oligomerization defines cell stiffness by influencing the actin architecture in renal epithelial cells. a** Representative SEM images of MDCK cells treated with DMSO (0.1%) or Bis-T-23 (30 µM, 0.1% DMSO) for 10 min prior to the addition of DMSO (0.1%) or LatA (0.2 µM, 0.1% DMSO) for 20 min. Enlarged images of the insets (orange boxed regions) show the arrangement, distribution, and density of microvilli at the apical membrane. **b** Representative PR-EM images of MDCK cells treated as explained in (**a**). The images show changes in the organization of the actomyosin cortex in MDCK cells under the indicated conditions. **c** Representative images of Young's Modulus maps of MDCK cells treated as explained in (**a**). **d**, **e** Bar graphs representing Young's Modulus depicting cell stiffness measured at the cell–cell junction (**d**) or at the apical membrane (**e**). Each symbol represents the average stiffness of a single cell. Results shown in **d**, **e** were generated from at least 10 cells from at least three culture dishes. Error bars, mean ± S.D. (*$P \leq 0.05$, **$P \leq 0.01$, ***$P \leq 0.001$, ****$P \leq 0.0001$, unpaired two-tailed $t$-test). ns, not significant.

---

**Table 1 Pharmacological stimulation of dynamin oligomerization counteracts drug-induced alterations at the apical membrane in MDCK cells.**

| Treatment | Cell height (µm) | Microvilli height (µm) | Microvilli per 9 µm² |
|---|---|---|---|
| Control | 11 ± 2.4 | 0.63 ± 0.2 | 53 ± 13 |
| DMSO (0.2%; 30 min) | 10 ± 2.5 | 0.42 ± 0.08 | 47 ± 18 |
| Bis-T-23 (30 µM; 30 min) | 13 ± 3.2 ($P < 0.001$) | 0.52 ± 0.07 ($P < 0.01$) | 55 ± 28 (ns) |
| LatA (0.2 µM; 30 min) | 3.4 ± 2.1 ($P < 0.0001$) | 0.39 ± 0.09 ($P < 0.001$) | 28 ± 11 ($P < 0.01$) |
| Bis-T-23→LatA (30 min) | 10 ± 2.8 ($P < 0.001$)* | 0.49 ± 0.12 ($P < 0.01$)* | 41 ± 8 ($P < 0.01$)* |
| DMSO (0.1%: 24 h) | 10.2 ± 2.4 | 0.42 ± 0.01 | 52 ± 8 |
| Bis-T-23 (5 µM; 24 h) | 11.6 ± 2.5 (ns) | 0.47 ± 0.08 (ns) | 53 ± 16 (ns) |
| DMSO→Cisplatin (24 h) | 5.2 ± 2.5 ($P < 0.0001$) | 0.18 ± 0.05 ($P < 0.0001$) | 19 ± 4 ($P < 0.0001$) |
| Bis-T-23→Cisplatin (24 h) | 9.6 ± 4.3 ($P < 0.001$)* | 0.34 ± 0.09 ($P < 0.001$)* | 40 ± 12 ($P < 0.0001$)* |

For LatA experiments, MDCK cells were treated with DMSO (0.1%) or Bis-T-23 (30 µM, 0.1% DMSO) for 10 min prior to the addition of DMSO (0.1%, vehicle) or LatA (0.2 µM, 0.1% DMSO) for 20 min. In the case of cisplatin experiments, cells were treated with DMSO (0.1%) or Bis-T-23 (5 µM, 0.1% DMSO) for 1 h, after which cisplatin (35 µM) was added for an additional 23 h. The cells were subsequently processed for SEM analysis. The data represent measurements of cell height ($n = 17$–30 cells per condition), microvilli height ($n = 33$–43 microvilli per condition), and microvilli density ($n = 11$–14 areas per condition) in MDCK cells. Statistical significance was determined between DMSO-treated cells and LatA, Bis-T-23, or cisplatin-treated cells, or between cisplatin- or LatA-only treated cells and those that were pre-treated with Bis-T-23 (*). Statistical significance was determined using an unpaired two-tailed $t$-test. Error bars, mean ± S.D.

---

We have also determined cell stiffness using the BioScope II system as an alternative experimental approach for AFM. In this instance, force indentation curves were obtained following the model of Discher and coworkers computed with Matlab software[23]. Similar trends with regard to cell stiffness were recorded for the interplay between LatA and Bis-T-23 (Supplementary Fig. 2b, 2c). In addition, dynole did not exhibit any effect on cell stiffness, whereas CytoD significantly decreased cell stiffness (Supplementary Fig. 2d, e), in accordance with their actin phenotypes (Supplementary Fig. 1a). Together, these data establish the correlation between the status of the actomyosin cortex, cell stiffness, the morphology of the apical membrane, and cell polarity. These findings also convincingly demonstrate the role of dynamin oligomerization in defining mechanical parameters of epithelial cell polarity via its effect on the actomyosin cortex.

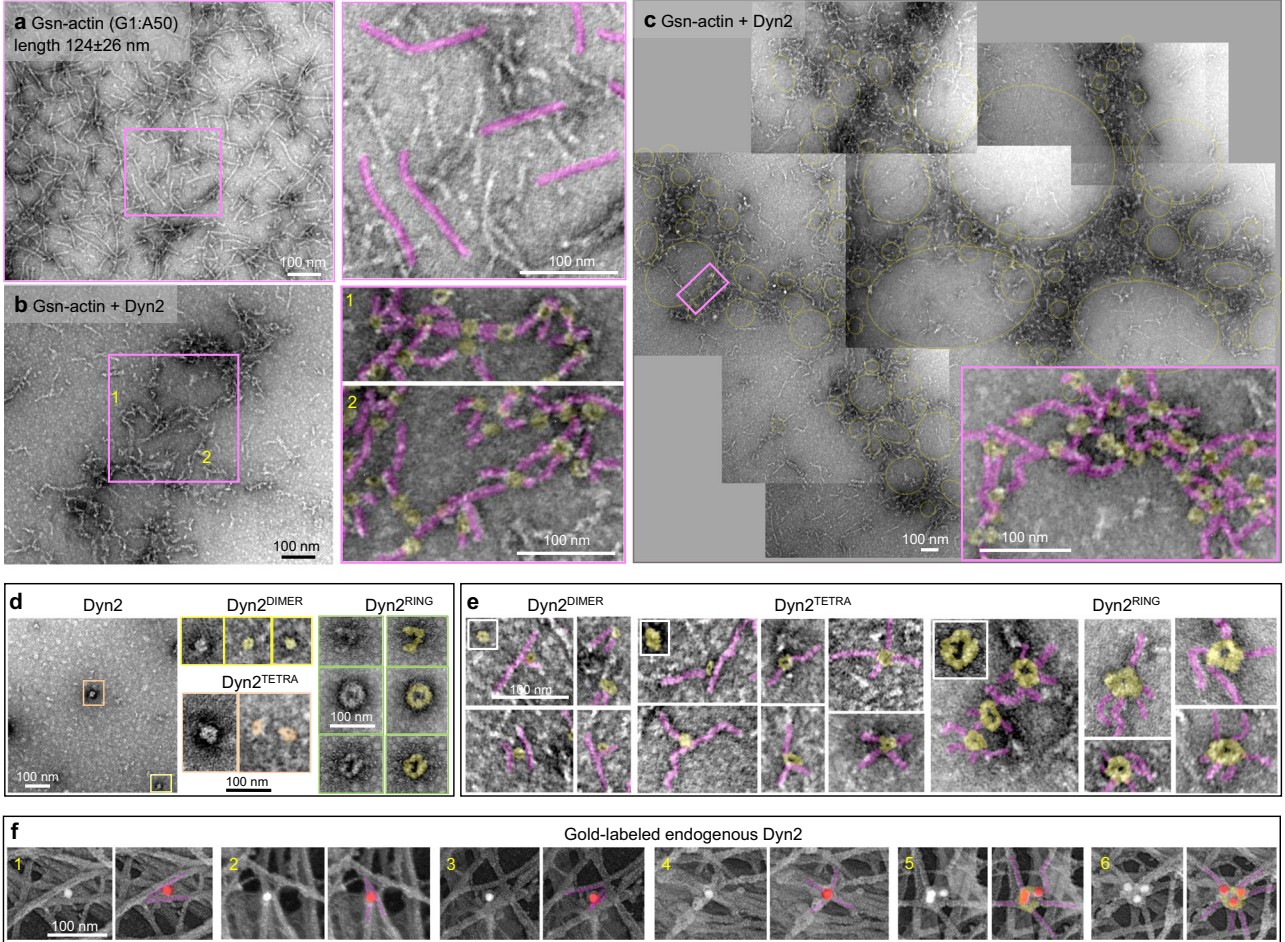

**Fig. 2 Dynamin cross-links actin filaments into branched networks. a–c** Representative TEM images of branched actin networks assembled by gelsolin-capped short actin filaments (Gsn-actin) and Dyn2. Gsn-actin (polymerized at the ratio of G1:A50) is shown in (**a**), and the actin networks assembled in the presence of Dyn2 are shown in (**b**). Several electron micrographs were stitched together to show the global organization of actin networks formed by Dyn2 in (**c**). A higher magnification image of the boxed region shows the structural arrangement of actin filaments (pink) and Dyn2 (yellow) within the networks. **d** Representative micrographs of negatively stained recombinant Dyn2. Boxed higher magnification images show distinct oligomerization states of dynamin: dimeric (yellow), tetrameric (orange), partial/full ring (green). **e** High magnification images of actin filaments (pink) associated with distinct Dyn2 oligomerization forms (yellow). Insets show images of only Dyn2 at different oligomerization states. **f** Immunogold platinum replica electron micrographs focusing on actomyosin cortex in MDCK cells. Cellular localization of endogenous Dyn2 was determined by monoclonal Dyn2 antibody and gold-conjugated secondary antibody (white particles). To better visualize the gold particles within tightly packed actin networks, white densities associated with gold particles were pseudo-colored red, and the actin filaments associated with gold particles were pseudo-colored pink. The presence of multiple gold particles identifies the formation of macromolecular dynamin complexes on actin filaments, consistent with the formation of dynamin rings. Representative images of two independent experiments.

**Dynamin cross-links actin filaments into branched networks that underlie cell polarity.** In order to elucidate the molecular mechanism by which dynamin oligomerization influences the architecture of the actomyosin cortex, we next examined the effect of dynamin on actin filaments in a reconstituted system. Based on the current hypothesis, the length of actin filaments defines their mode of cross-linking[6]. As the average length of cortical actin filaments within a network at the leading edge is between 100 and 150 nm[24], we examined the effects of dynamin on the organization of shorter filaments generated by capping F-actin with gelsolin (Gsn-actin) (Fig. 2a). The addition of Dyn2 resulted in the formation of large, branched networks (Fig. 2b, c). Based on the sizes and shapes of recombinant Dyn2 (Fig. 2d), the networks were formed predominantly by Dyn2 dimers (Dyn2DIMER) and tetramers (Dyn2TETRA) that interacted with several actin filaments (Fig. 2e): Dyn2DIMER bound up to two filaments, Dyn2TETRA bound up to four filaments, and Dyn2RING bound up to six filaments. Low magnification of the images revealed that dynamin-

dependent networks form a pattern of smaller and larger ring-like shapes (Fig. 2c).

To correlate observations from the reconstituted system with dynamin's role in cells, we next determined the localization of endogenous Dyn2 on cortical actin networks using a monoclonal anti-Dyn2 antibody followed by a gold-conjugated secondary antibody (Supplementary Fig. 3a–c). As seen in the reconstituted system, dynamin associated with a distinct number of F-actin within branched networks (Fig. 2f). Together, these data identify a novel activity of dynamin, that is cross-linking F-actin into branched networks.

To correlate dynamin's cross-linking capability and the protective effect of Bis-T-23 on the actomyosin cortex and the morphology of the apical membrane, we next examined the effects of Bis-T-23 on dynamin-mediated networks in reconstituted systems (Fig. 3a). Based on contour plots, which provide topographical representations of varying filament densities, Bis-T-23 increased overall network density (Fig. 3a), which could be

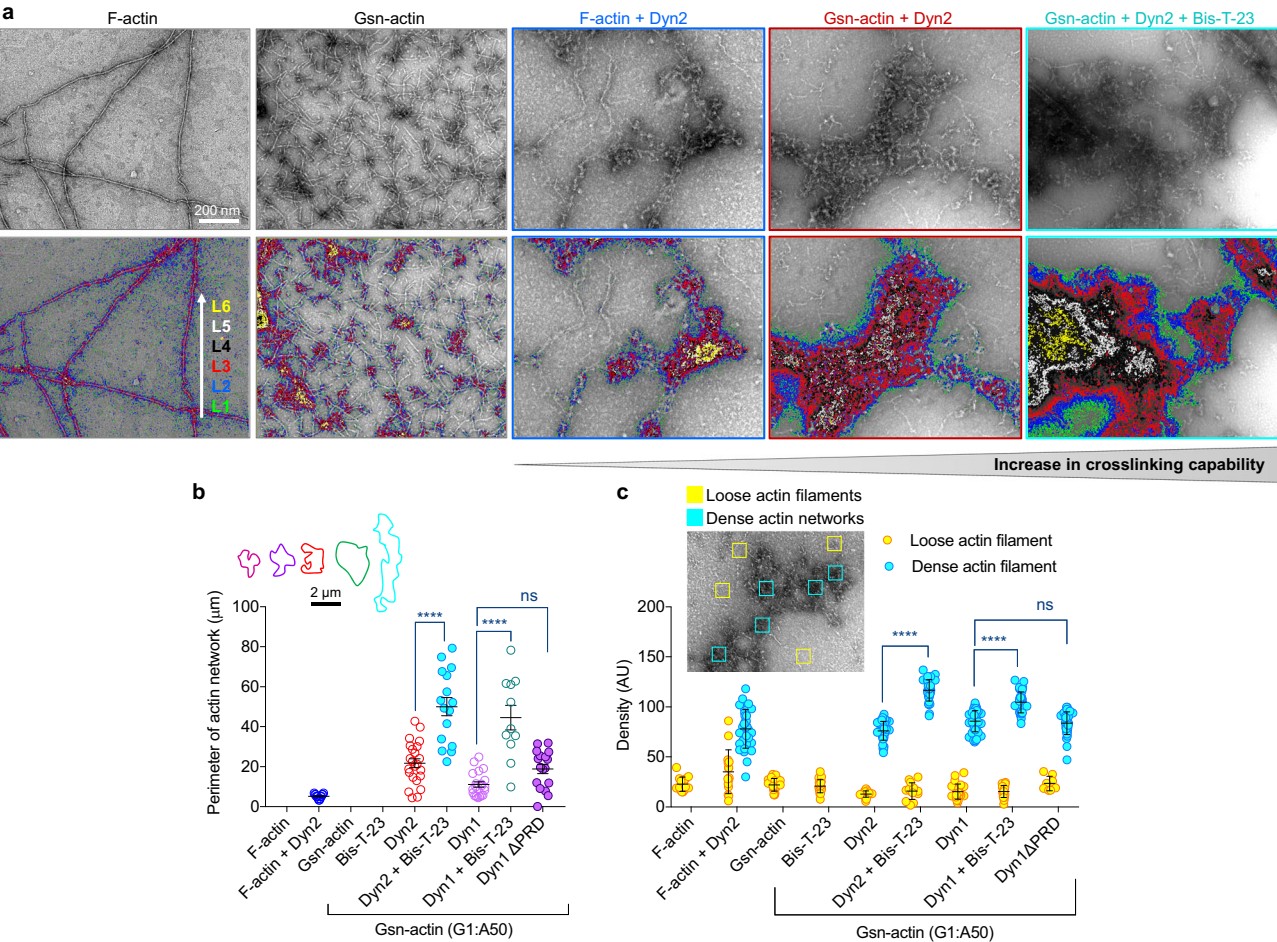

**Fig. 3 Dynamin oligomerization status and the length of actin filaments define its cross-linking capability. a** Electron micrograph of actin arrays assembled from F-actin or gelsolin-capped actin filaments (Gsn-actin) by Dyn2 in the presence or absence of Bis-T-23 (0.4 μM). The lower panel shows contour plots depicting density as varying depths within a network. Distinct levels of thickness were color-coded as indicated in the figure. Contour plots were produced based on original EM images in the upper panel. **b** Bar graph illustrating the size of actin networks shown in (**a**). The inset shows the representative size (parameter) of actin networks for each condition. 11 to 23 actin networks were analyzed. **c** Graph depicting the density of the actin filaments incorporated into actin networks (blue) or loosely distributed through the background of the micrograph (yellow). Data were generated by measuring density within a defined square (100 × 100 nm) as shown in the representative insert. Results shown were generated based on EM data shown in (**a**). 33–45 dense networks (blue squares) and 14–27 loose networks (yellow squares) were counted per condition. Data in (**b**), (**c**) are plotted as mean ± S.E.M. and mean ± S.D, respectively (****$P < 0.0001$, one-way ANOVA with Tukey's multiple comparison test). ns, not significant.

explained by the increase in the number of F-actin bound to dynamin due to the increase in its oligomerization. In addition, dynamin more potently cross-linked shorter filaments than long F-actin (Fig. 3a–c), suggesting that dynamin's capability to form branched networks is defined by its oligomerization status and the length of actin filaments. The ability to cross-link actin filaments into networks was shared by two dynamin isoforms, ubiquitously expressed Dyn2 and neuron-specific dynamin-1 (Dyn1) (Fig. 3b, c).

We and others have shown that dynamin binds F-actin via two distinct domains: one situated in the Middle domain[10,12,25–27], and one part of the C-terminal proline-arginine rich domain (PRD)[28]. Similar networks were formed by Dyn1 and Dyn1$^{\Delta PRD}$ (Supplementary Fig. 3d), demonstrating that PRD is dispensable for network formation. Finally, Dyn2 also cross-linked F-actin into tight bundles and hyper-bundles (Supplementary Fig. 4a, 4b), as seen before[10,28]. In contrast to network formation, dynamin's ability to form bundles is dependent on PRD-actin interactions[28]. Bis-T-23 promoted the formation of hyper-bundles (Supplementary Fig. 4c, 4d), suggesting a mechanism by which Bis-T-23 increased the number of microvilli on the apical membrane

(Table 1). This hypothesis is supported by the presence of anti-dynamin antibodies within actin bundles that underlie filopodia (Supplementary Fig. 4e). Together, these data show that dynamin's ability to cross-link filaments into networks and bundles is influenced by its propensity to oligomerize as well as the length of actin filaments (Supplementary Fig. 4f).

Insights generated by the reconstituted system were tested in the cell-based assays. Expression of both dynamin isoforms lacking PRD (Dyn$^{\Delta PRD}$) in MDCK cells did not affect the ZO-1 staining pattern when compared to wild-type dynamins, and Dyn2$^{\Delta PRD}$ also increased F-actin levels (Supplementary Fig. 5a). These data suggested that dynamin's ability to form tight bundles does not contribute to the overall level of F-actin or the cell polarity in MDCK cells. In contrast, expression of Dyn2$^{K/E}$, mutant impaired in F-actin binding[10], decreased F-actin levels and cell polarity compared to Dyn2 expressing cells (Supplementary Fig. 5a–c), implicating the Middle domain in the formation of actomyosin networks that underlie cell polarity. This hypothesis is further supported by AFM experiments showing that Dyn2$^{K/E}$ also decreased cell stiffness at the apical membrane when compared to Dyn1$^{WT}$ (Supplementary Fig. 5d).

These studies are complementary to experiments using mouse inner medullary collecting duct (mIMCD) cells, another cell line representing polarized renal epithelial cells. In Dyn2 knockdown mIMCD cells, a significant decrease in F-actin levels and a patchy ZO-1-staining pattern were observed (Supplementary Fig. 6a–c). The F-actin phenotype was rescued to normal level by the expression of either wild type or Dyn2$^{\Delta PRD}$, but not by Bis-T-23 (Supplementary Fig. 6d–g). Together these data provide evidence for the essential role of dynamin and its oligomerization in the establishment and maintenance of epithelial cell polarity. They also strongly suggest that actin networks that underlie the apical membrane are formed by interactions between F-actin and the Middle domain, and that dynamin's ability to form bundles and hyper-bundles is not essential for cell polarity.

**Pharmacological stabilization of cortical actin via dynamin ameliorates AKI.** Loss of renal tubules brush border due to disassembly of actomyosin cortex is a predominant feature of AKI[2,29]. A widely used anti-cancer drug often leading to AKI is cisplatin[30]. A pivotal consequence of cisplatin injury in renal cells is an increase in reactive oxygen species (ROS) level[31]. Moreover, increased levels of ROS lead to a decrease in actin dynamics[32], which causes disruption of the mitochondrial membrane; this further exacerbates ROS production[33,34]. Thus, we examined whether pharmacological stimulation of dynamin's cross-linking capability is sufficient to counteract cisplatin-induced injury.

As seen before[35], cisplatin decreased MDCK cell height, microvilli number, and length, and affected cell polarity without affecting endocytosis (Table 1, Supplementary Fig. 7a–d). TEM analysis suggested that these phenotypes could be attributed to cisplatin-induced loss of F-actin (Fig. 4a). The addition of Bis-T-23 prior to cisplatin abrogated its effects by stabilizing the actomyosin cortex against cisplatin-induced injury (Fig. 4a–c). Similar phenotypes were observed in porcine kidney proximal tubule (LLC-PK1) cells (Supplementary Fig. 8). Together, these data provide evidence that Bis-T-23 preserves the actomyosin cortex from cisplatin-induced disassembly.

In a complementary physiologically relevant approach[36], Bis-T-23 counteracted cisplatin's ability to reduce transepithelial electrical resistance (TER) as measured across the live epithelial monolayer using a transwell culture system (Fig. 4d). Since TER measures the integrity of tight junctions in cell culture, these data further establish the beneficial effect of Bis-T-23 on epithelial cell integrity upon injury.

As altered actin dynamics can further enhance ROS production, we next examined whether stabilization of the actin network by Bis-T-23 can counteract oxidative stress in cisplatin-injured cells. We elected to assess the status of cellular biomolecule carbonyls, a stable biomarker of oxidative damage produced by high levels of ROS, using our recently developed fluorescent sensor TFCH[37] (Fig. 4e). Indeed, the addition of Bis-T-23 prior to cisplatin partially diminished cisplatin-induced increase in carbonylation (Fig. 4f). Similar trends regarding carbonylation were observed after treatment with Bis-T-23 and the actin depolymerizer LatA (Supplementary Fig. 9a). Together these data provide evidence that stabilization of the actin networks can counteract the feedback loop between oxidative stress and the status of actin in injured cells.

An ex vivo rat kidney model was next utilized to visualize Bis-T-23-driven preservation of the actomyosin cortex at the brush border. Cisplatin induced a significant loss of F-actin staining at the brush border of proximal tubules (Fig. 4g). A similar loss of F-actin at the brush border upon the onset of severe ischemia-reperfusion injury was recently observed using intravital

imaging[38]. Pre-treatment with Bis-T-23 partially protected cisplatin-induced loss of F-actin at the brush border of renal tubules (Fig. 4g), further demonstrating the effect of Bis-T-23 on actomyosin cortex at the apical membrane of renal epithelial cells. As seen in the cell culture, Bis-T-23 did not affect endocytosis in the tissue slices (Supplementary Fig. 9b), further demonstrating that Bis-T-23 does not affect dynamin's role in endocytosis in renal proximal tubules.

Finally, we examined the reno-protective effect of Bis-T-23 in a mouse model of cisplatin-induced AKI[39]. BL6 mice were injected with either Bis-T-23 or DMSO once per day starting 24 h prior to injection of cisplatin. As seen before[40], serum creatinine (SCr) and blood urine nitrogen (BUN) levels significantly increased 3 days post cisplatin injection (Fig. 4h). DMSO exhibited mild reno-protection as seen before[41], due to its oxidant scavenging ability[42]. Daily administration of Bis-T-23 outperformed DMSO with regard to reno-protection (Fig. 4h). As all animals were sacrificed on day 5 due to the systemic toxicity of cisplatin (Supplementary Fig. 9c), we could not examine the long-term benefits of Bis-T-23 on kidney function. Together, these studies persuasively demonstrate that preserving renal cell integrity via stabilization of the actomyosin cortex at the apical membrane counteracts cisplatin-induced nephrotoxicity.

**Stabilization of the actin networks counteracts iohexol-induced AKI.** While the exact mechanism of contrast dye-induced AKI is not fully deciphered, it remains a prime cause of hospital-acquired AKI[43]. Physiologically, contrast dye increases the osmotic load, decreases renal blood flow, and induces renal arterial constriction[44]. At the cellular level, ROS is a critical player in contrast to dye-induced AKI[45]. Given the role of ROS in regulating the actin cytoskeleton and our hypothesis that stabilization of the actomyosin cortex can counteract oxidative damage, we examined the effects of Bis-T-23 on a contrast dye-, iohexol-induced AKI[46].

The cellular effects of iohexol were examined in human kidney proximal tubular (HK-2) cells. As seen for cisplatin, iohexol induced loss of F-actin, which was accompanied by a significant increase in the level of carbonylated molecules, an irreversible consequence of elevated ROS levels (Fig. 5a, b). Both phenotypes were partially counteracted by pre-treatment with Bis-T-23, further establishing the synergy between the status of the actin cytoskeleton and the level of oxidative stress in these cells. Complementary to cell-based assays, daily administration of Bis-T-23 preserved renal function based on the Scr and BUN levels in BL6 mice challenged with contrast dye (Fig. 5c). A much-coveted response of improved survival rate was also observed in the animals that received Bis-T-23 along with the contrast dye (Fig. 5d).

We have recently shown that elevated serum levels of soluble urokinase-type plasminogen activator receptor (suPAR) sensitize the kidney to iohexol-induced AKI[47]. We next investigated if dynamin-mediated protection of the tubular cells can also render reno-protection in BL6 mice expressing high levels of suPAR from fat tissue (suPAR-Tg)[48]. As expected[47], the suPAR-Tg mice exhibited a decrease in kidney function 24–48 h post iohexol injection (Fig. 5e). In contrast, the animals that received a daily dose of Bis-T-23 exhibited significantly better kidney function despite the contrast dye challenge (Fig. 5e). Notably, a significant decrease in the mortality rate was observed in the mice cohort that received Bis-T-23 (Fig. 5f). As anticipated, the extent of oxidative damage (biomolecule carbonylation) in HK-2 cells concomitantly treated with Bis-T-23 was significantly lower than in cells that received only a combination of suPAR and Iohexol (Supplementary Fig. 9d).

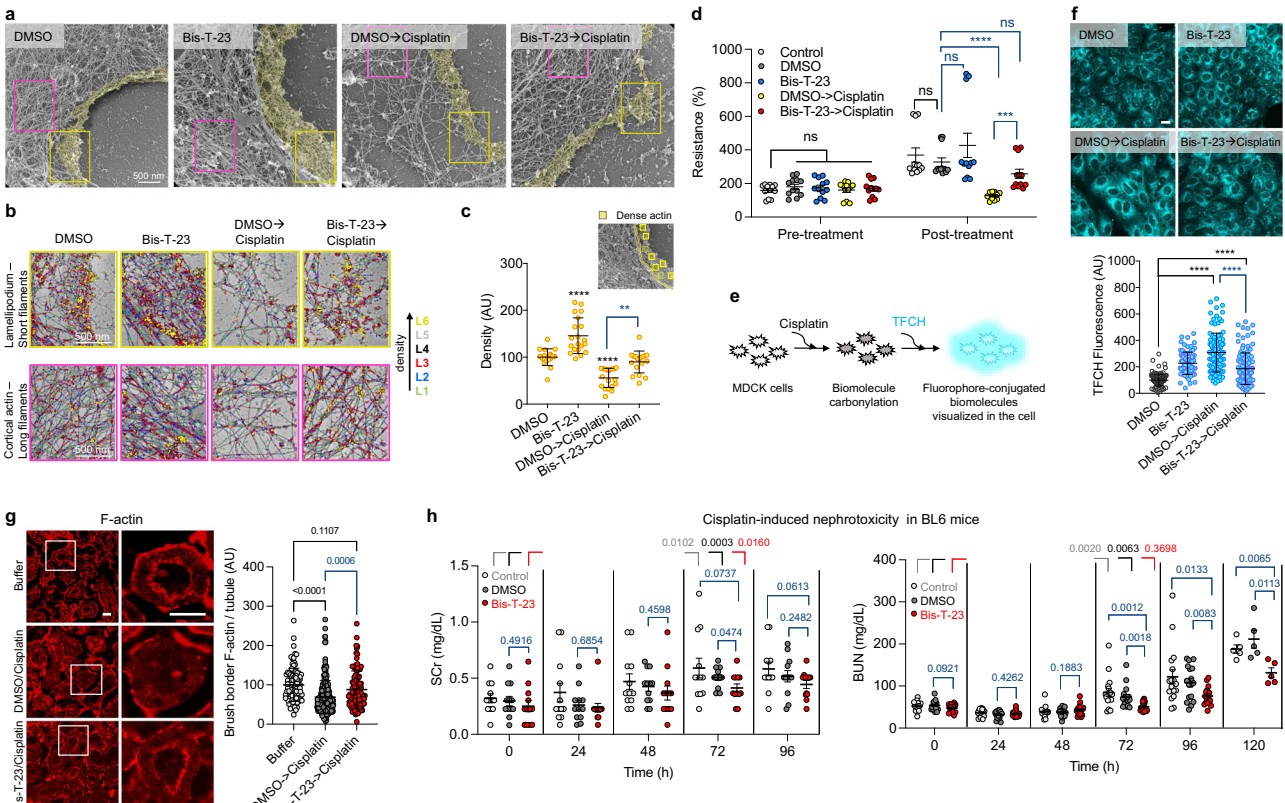

**Fig. 4 Dynamin agonist protects renal tubules from cisplatin-induced injury by stabilizing the actomyosin cortex at the apical membrane. a** PR-EM image of lamellipodium in MDCK cells treated with DMSO (0.1%) or Bis-T-23 (5 μM, 0.1% DMSO) for 1 h prior to the addition of cisplatin (35 μM) for 23 h. The actomyosin cortex was colored yellow. **b** Higher magnification images of boxed regions in (**a**). Distinct levels of thickness are color-coded. The upper and lower panels show networks formed by shorter filaments and longer filaments, respectively. **c** Graph depicting the density of the actin filaments at the leading edge (yellow box in (**a**); 18–20 areas counted). **d** Bar graph depicting relative transepithelial electrical resistance (%) in live MDCK cells treated as stated in (**a**) except for cisplatin (50 μM) and Bis-T-23 (30 μM) concentrations (12 readings per sample). Data are plotted as mean ± S.E.M. (***$P$ < 0.001, ****$P$ < 0.0001, unpaired two-tailed $t$-test). ns, not significant. **e** Schematic representation of oxidative stress-induced carbonylation detection using TFCH assay. **f** Images showing the levels of biomolecule carbonyls determined by TFCH. Cells were treated as described in (**a**) except for cisplatin concentration (5 μM). Scale bar, 20 μm. Graph depicting the relative levels of TFCH fluorescence associated with biomolecule carbonyls per cell (data points ($n$) = 83–116; each point represents average intensity of 4–9 cells). **g** Rat kidney slices stained with anti-actin antibody. Kidney slices were incubated in buffer, DMSO (0.1%), or Bis-T-23 (30 μM) for 1 h before adding cisplatin (200 μM) or buffer for 8 h. White squared regions were enlarged. Scale bars, 40 μm. Graph showing F-actin intensity at the brush border per tubule (100-236). For **c**, **f**, **g**, data are plotted as mean ± S.D. ($P$ values are reported in the Figure, one-way ANOVA with Tukey's multiple comparison test). **h** Scatter dot plots showing AKI induced by cisplatin determined by the level of blood urea nitrogen (BUN) or serum creatinine (SCr). Animals were injected with either DMSO (1%) or Bis-T-23 (20 mg/kg) ($n$ = 12, per condition for SCr; $n$ = 17, per condition for BUN) once a day starting 24 h prior to cisplatin (15 mg/kg) injection. The measurements were performed at the indicated times. Data are plotted as mean ± S.E.M. ($P$ values are reported in the Figure, unpaired two-tailed $t$-test). ns, not significant.

In contrast to ROS-mediated cellular injury caused by cisplatin and iohexol, suPAR promotes tubular cell injury partly by increasing oxygen consumption rate (OCR)[47]. To link the physiological effect of Bis-T-23 in suPAR transgenic mice to its effect on cell metabolism, we examined whether stabilization of the actin network via dynamin decreases suPAR-driven OCR. OCR was measured in real-time in HK-2 cells under basal conditions and in response to sequential injections of mitochondrial inhibitors using a Seahorse XFe24 extracellular flux analyzer (Fig. 5g). The addition of an anti-suPAR antibody or Bis-T-23 ameliorated suPAR-driven increases in mitochondrial basal respiration, ATP production, maximum rate of respiration, and spare-respiratory capacity (Fig. 5f). These data demonstrate the positive effects of Bis-T-23 on cell metabolism upon injury. Together, these studies further affirm the feasibility of protecting against multiple types of AKI and thereby improving the survival rate in rodent models via dynamin as a proxy.

**Simultaneous stabilization of actin in distinct kidney cells attenuates nephron injury.** We have previously shown that pharmacological activation of dynamin-driven actin polymerization restored the structure and function of podocytes in diverse murine models of CKD[12]. In this study, we demonstrated dynamin's ability to preserve the integrity of tubular cells upon acute injury by cross-linking the actin cytoskeleton. Together, these insights led us to envision the possibility of simultaneously counteracting both, glomerular and tubular injury, by pharmacologically targeting dynamin.

To test this hypothesis, we examined whether Bis-T-23 could delay the loss of kidney function in the mouse model of Alport syndrome (AS). AS is an inherited form of progressive kidney failure due to mutations in the *COL4A3, COL4A4,* or *COL4A5* genes that together encode type IV collagen, a major component of the glomerular basement membrane (GBM)[49,50]. Although the primary defect in AS is foot process effacement and proteinuria

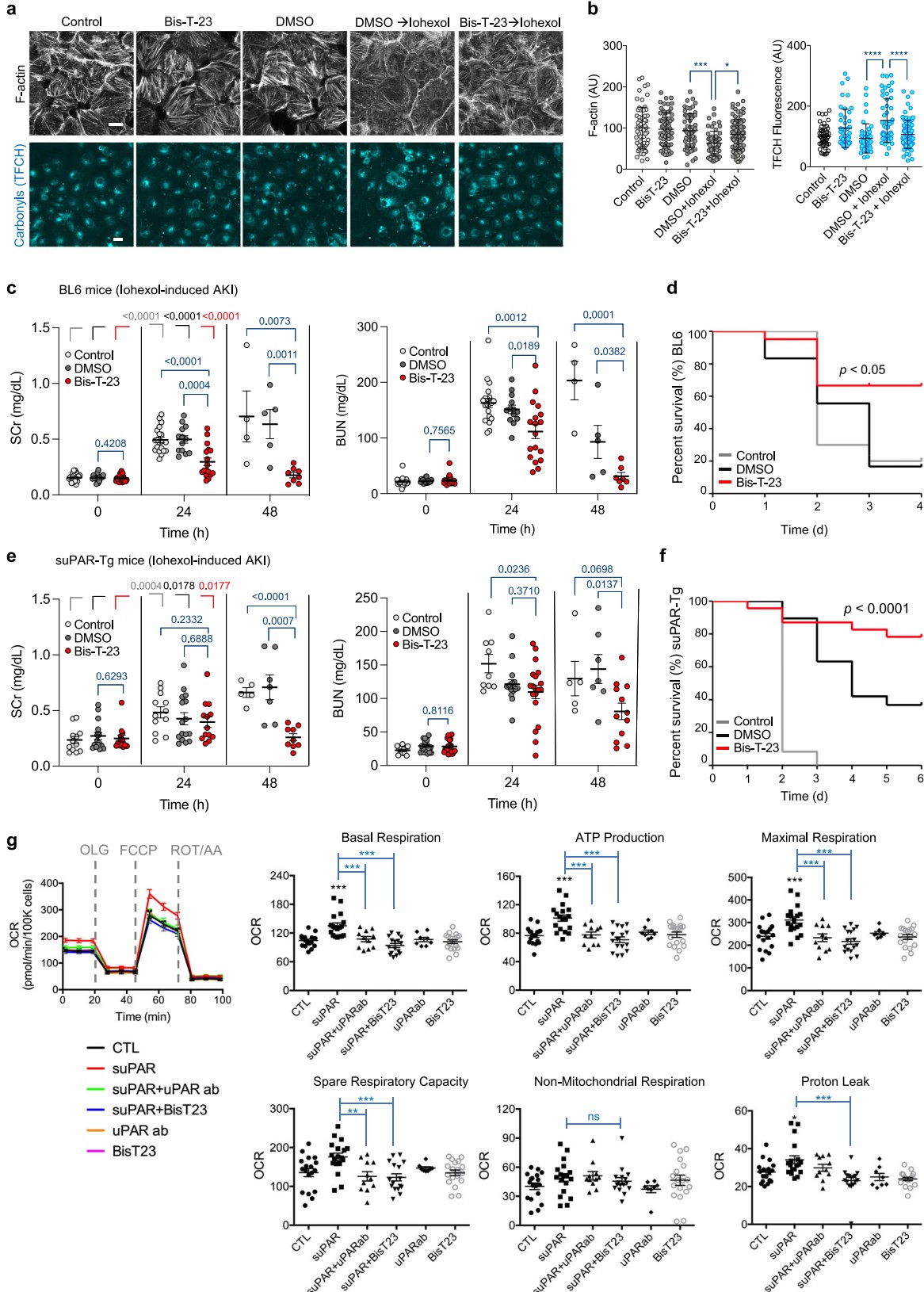

due to alteration in the composition of the GBM, the tubular response is also critical to the pathogenesis of AS and may precede the deterioration of glomerular function[51–53]. Indeed, signaling between tubules and glomerulus influences the response of the podocytes to collagen deficiency[54]. Currently, there is no specific therapy for AS.

Daily administration of Bis-T-23 slowed down the increase in proteinuria in *Col4a3*[−/−] animals[55] (Fig. 6a), demonstrating that dynamin-mediated increase in actin polymerization in podocytes can counteract even genetic defects in GBM. Bis-T-23 also slowed down the decrease in kidney function, indicated by a small increase in BUN levels, protected tubules from developing hyaline

**Fig. 5 Dynamin agonist counteracts Iohexol-induced AKI. a** Status of F-actin and biomolecule carbonylation in HK-2 cells. HK-2 cells were treated with medium (Control), DMSO (0.1%), or Bis-T-23 (10 μM, 0.1% DMSO) for 1 h. The medium was replaced with only medium or medium containing Iohexol +DMSO (250 mg/mL (F-actin set) and 100 mg/mL (TFCH assay)) or iohexol+Bis-T-23 for ~3 h. Scale bar, 20 μm. **b** Graphs depicting the relative levels of F-actin within a fixed region of interest (ROI) in the cells (58–98 ROI analyzed) TFCH fluorescence (cellular biomolecule carbonyls) (data points (n) = 46–83; each point represents average intensity of 1–4 cells). Error bars, mean ± S.D. (*$P < 0.05$, ***$P < 0.001$ ****$P < 0.0001$, one-way ANOVA with Tukey's multiple comparison test). **c, e** Scatter dot plots showing AKI induced by Iohexol (5 g/kg) determined by the level of SCr or BUN in BL6 wild-type mice (**c, d**) or suPAR-Tg mice (**e, f**). **c, e** Error bars, mean ± S.E.M. (P values are reported in the Figure, unpaired two-tailed t-test). As indicated, animals were injected with either DMSO (1%) or Bis-T-23 (20 mg/kg) once a day starting at 0 h. The measurements were performed at the indicated times. Number of animals per condition in **c**: BL6 control (n = 20), DMSO (n = 18), and Bis-T-23 (n = 21). Number of animals per condition in (**e**): suPAR-Tg control (n = 12 for BUN or SCr), DMSO (n = 19 for BUN; n = 16 for SCr), and Bis-T-23 (n = 20 for BUN; n = 13 for Scr). **d, f** Log-Rank (Mantel-cox) test was performed to analyze survival curves of animals treated as described in (**c**), (**e**). P was calculated using an unpaired two-tailed t-test. **g** Oxygen consumption rate (OCR) curves and Seahorse XF analyzer measurements of mitochondrial respiration of HK-2 cells. HK-2 cells were treated with suPAR (10 ng/ml), human anti-uPAR antibody (50 ng/ml), or Bis-T-23 (10 μM) alone or in combination for 24 h. OCR was measured in real-time under basal conditions and in response to sequential injections of mitochondrial inhibitors including oligomycin (OLG; an ATP synthase inhibitor), FCCP (an uncoupler of ATP synthesis from oxygen consumption), rotenone (ROT; complex I inhibitor), and antimycin A (AA; complex III inhibitor). Each OCR value was normalized to cell number and is presented as pm/min/100,000 cells. Graphs represent three independent experiments and are derived from the mean values (S.E.M.) of at least 8 replicates per group. Error bars, mean ± S.E.M. (**$P < 0.01$, ***$P < 0.001$, one-way ANOVA). ns, not significant.

casts, and extended the lifespan of animals (Fig. 6a–c). These data further attest to the positive impact of increased dynamin cross-linking activity on the preservation of renal tubular cell integrity. Together, our study demonstrates the suitability and the unique advantage of pharmacological targeting of nephron as a unit via dynamin as a proxy (Fig. 6d).

## Discussion

Since the identification of direct dynamin–actin interactions[10], a growing body of evidence is establishing this GTPase as one of the major regulators of the actin cytoskeleton in the cell. In contrast to canonical ABPs, the mechanisms by which dynamin influences actin are both, highly versatile and cell-type specific. This is due to the combination of dynamin's multiple oligomerization states and its ability to bind F-actin by two different binding sites. F-actin interactions with dynamin rings and helices via its C-terminal PRD result in actin bundles and hyper-bundles, which have been implicated in the formation of filopodia[56] and membrane protrusions during myoblast fusion[28]. Interactions between dynamin's Middle domain and F-actin have been implicated in regulating actin networks in lamellipodia[26], postsynaptic cytoskeleton organization, and neuromuscular junction development[27], actin bundle rigidity in invadosomes during myoblast fusion[25], and formation of podocyte foot processes[12]. While the formation of foot processes is driven by dynamin-stimulated actin polymerization, this study suggests that the molecular mechanism by which dynamin influences these other actin-driven processes might be in part due to its ability to cross-link F-actin into branched networks. Our current study expands the role of dynamin-dependent network formation to include formation of actomyosin cortex at the apical membrane of polarized epithelial cells and the establishment of cell stiffness.

Despite the mechanistic diversity, dynamin's capability for cross-linking F-actin or promoting actin polymerization is enhanced by the increase in its oligomerization state (this study and refs. [12,28]). Dynamin oligomerization is cooperative and is regulated by its concentration[57], the length of actin filaments (this study and ref. [13]), and SH3-domain-containing proteins[25,58]. The combination of all these mechanisms ultimately defines the temporal, spatial, and cell-type specificity of dynamin's role in modifying and/or establishing diverse actin structures.

Changes in mitochondrial function and cell metabolism have been linked to a multitude of AKI etiologies[59]. Mitochondria are dynamic organelles that respond to physiological signals and are significant sources of ROS in healthy[60] and injured cells[61]. Cisplatin and iohexol directly damage mitochondria leading to

increased production of ROS[37,43,62–64], while suPAR affects the bioenergetic parameters of renal cells[47]. As many cytoskeletal proteins are sensitive to ROS[32,65], a decrease in actin dynamics in the presence of elevated levels of ROS has been reported[32]. Meanwhile, the dysregulated actin cytoskeleton further augments ROS production[33], suggesting a feedback loop between oxidative stress and the actin cytoskeleton. Here we show, by using a combination of cell-based assays focusing on cell polarity, real-time extracellular flux experiments, and our new carbonylation-specific fluorophore, that stabilization of actomyosin cortex via dynamin partially attenuates the feedback loop between oxidative stress and actin cytoskeleton dynamics regardless of the initial type of injury (cisplatin, LatA, iohexol, or suPAR).

It is well established that CKD and AKI, though mechanistically distinct, are closely interconnected, with AKI being recognized as a risk factor for CKD development and progression[66]. As of now, AKI remains undruggable[67]. This is particularly relevant in light of the current coronavirus disease 2019 (COVID-19) pandemic, where hospitalized patients with COVID-19 and an elevated suPAR plasma level develop AKI at alarming rates[68]. Elevated suPAR levels have been linked to both AKI[47] and CKD[69], further reinforcing the molecular link between these two distinct kidney diseases. Therefore, it is becoming increasingly essential to develop therapeutics that can protect against an array of renal insults on multiple cell types[70]. Here we report the reno-protective effect of a dynamin agonist in a genetic model of AS, which exhibits injury to both podocytes and tubular cells. Our study establishes a comprehensive approach for developing novel therapeutics that engage with the actin network in multiple cell types within the nephron and treat myriad kidney diseases regardless of the site of injury.

## Methods

**Cell culture**. MDCK (Madin-Darby canine kidney) cells (ATCC CCL-34) and mouse inner medullary collecting duct (mIMCD-3) cells (ATCC CRL-2123) were grown in DMEM/F12 (ThermoFisher Scientific) containing 10% fetal bovine serum (FBS) and 1X Antibiotic-Antimycotic (penicillin, streptomycin, and Amphtericin B; Gibco). LLC-PK1 (Lilly Laboratories Cell-Porcine Kidney 1) cells (ATCC CL-101) were grown in DMEM with 4.5 g/L glucose, L-glutamine, sodium pyruvate (Corning), 10% FBS, and 1X Antibiotic-Antimycotic. MDCK and LLC-PK1 cells were grown for ~24 h and ~72 h, respectively, before initiating experiments. HK-2 (Human Kidney 2) proximal tubular cells (ATCC CRL-2190) were cultured in DMEM/Ham's F12 medium (Corning) containing 10% FBS, 100 U/ml penicillin, and 100 μg/ml streptomycin when used for bioenergetic experiments. For analyzing the status of actin cytoskeleton and cellular carbonylation, HK-2 cells were grown in DMEM F-12 (ATCC) containing 10% FBS, 1X Antibiotic-Antimycotic, and 1X ITS (insulin, transferrin, and sodium selenite) liquid media supplement (Sigma-Aldrich). To establish dynamin 2 (Dyn2) knock-down cell line (mIMCD-Dyn2^KD), cells were infected with lentiviruses carrying shRNA encoding *DNM2*

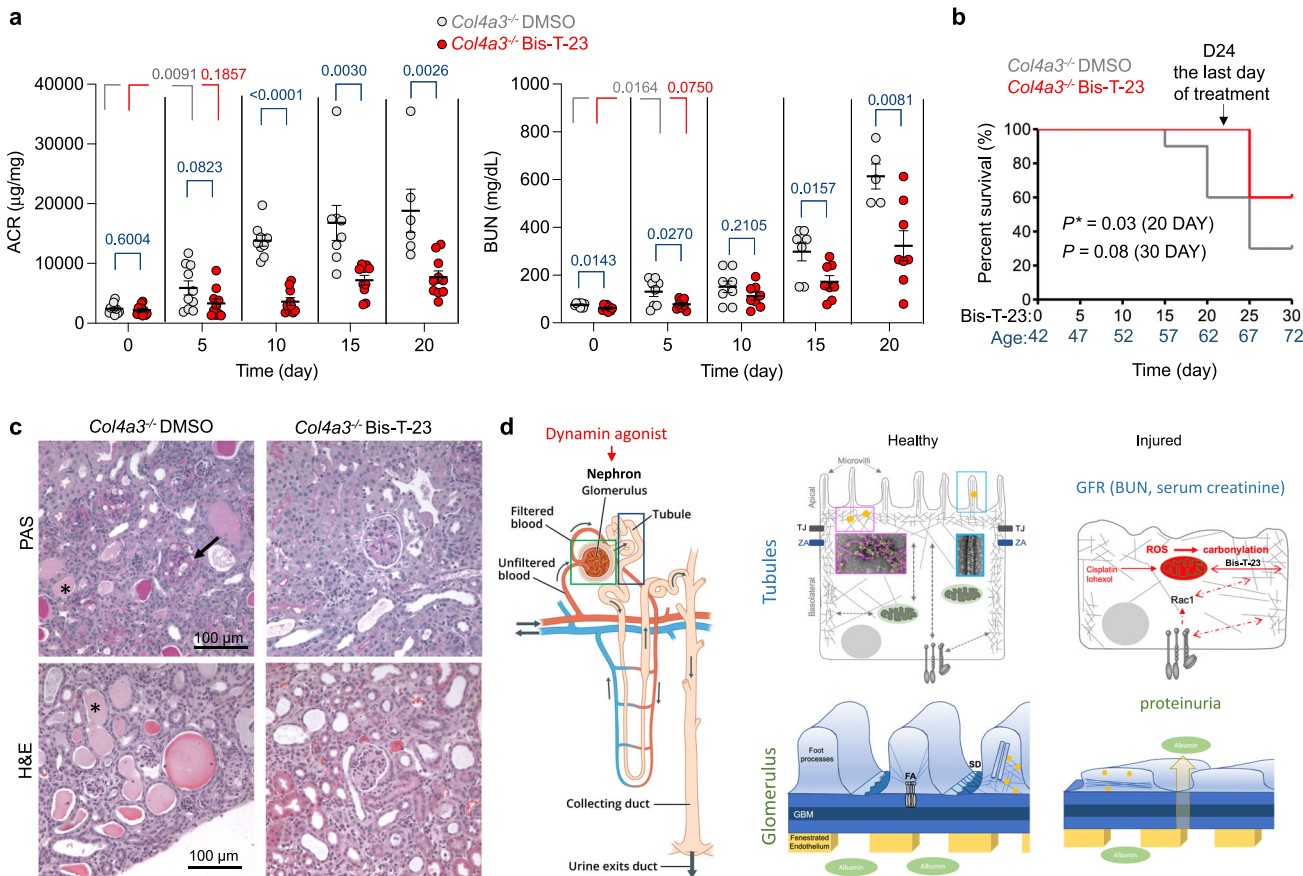

**Fig. 6 Pharmacological activation of dynamin simultaneously targets the actin cytoskeleton in two distinct cells of the nephron. a** $Col4a3^{-/-}$ mice were injected with DMSO (1%) or Bis-T-23 (20 mg/kg) once a day when 42 days old (Day 0). An equal number of DMSO or Bis-T-23 treated mice ($n = 10$ for ACR; $n = 8$ for BUN) was used. The scatter dot plot shows the level of proteinuria (albumin to creatinine ratio, ACR) and kidney injury determined by the level of BUN. Error bars, mean ± S.D. (P values are reported in the Figure, unpaired two-tailed t-test). **b** Log-Rank (Mantel-cox) test was performed to analyze the survival curves of the two treatment groups. **c** Representative histopathology of kidney determined by periodic acid-Schiff (PAS) and H&E staining on samples obtained from $Col4a3^{-/-}$ mice described in (**a**). Animals were 62 days old (day 20 of Bis-T-23 or DMSO treatment). Glomerulopathy in $Col4a3^{-/-}$ DMSO-treated animals was characterized by mesangial expansion observed on the PAS staining (black arrow), increased cellularity, and an increase in Bowman's space. Degenerative changes in tubules included the formation of casts (*) and the presence of tubular sclerosis. Bis-T-23 treatment over 20 days partially preserved the morphology of glomeruli (diminished mesangial expansion) and tubules (absence of casts). **d** Schematics of the nephron (image credited to National Institute of Diabetes and Digestive and Kidney Diseases, National Institutes of Health) and the mechanism involved in dynamin regulated reno-protection. In polarized epithelial cells of renal tubules, dynamin establishes cell polarity by cross-linking actin filaments into networks (this study). Dynamin's ability to cross-link filaments into bundles is implicated in the formation of microvilli. Acute injury increases ROS production, alters cell metabolism, and perturbs the actin dynamics, which further enhances ROS production leading to more pronounced cellular oxidative damage (biomolecule carbonylation). An increase in dynamin's cross-linking capability in the actomyosin cortex preserves tubular cell integrity, which partially protects them from oxidative insult. Additionally, dynamin is essential for the structure and function of podocytes. Pharmacological activation of dynamin restores foot processes by inducing actin polymerization. Our study shows that pharmacological activation of dynamin oligomerization exhibits a dual beneficial effect on the nephron by targeting two distinct molecular mechanisms: preservation of the integrity of podocytes (actin polymerization) and renal tubular cells (cross-linking of actin filaments).

gene (5′CGGCCTAGTGGACATGACAATGAACTCGAGTTCATTGTCATGTC CACTAGGTTTTTG3′)[10] or scrambled shRNA (Sigma-Aldrich) in complete media containing polybrene (8 μg/ml). After 24 h, the media was replaced every other day with media containing puromycin to select for stable transfectants. The extent of Dyn2 knock-down was assessed by western blot. All cell lines were grown on collagen-coated glass coverslips.

**Primary antibodies.** Mouse monoclonal anti-Dyn1/2 (Hudy 1) (Sigma-Aldrich MABT188); mouse monoclonal anti-dynamin 2 (Sigma-Aldrich SAB4200661); rabbit polyclonal anti-Dyn2 (ThermoFisher Scientific PA5-19800); rabbit polyclonal anti-ZO-1 (ThermoFisher Scientific 61-7300); mouse monoclonal anti-actin antibodies (Sigma-Aldrich A4700); human anti-uPAR antibody (R&D Systems MAB807).

**Secondary antibodies.** Rabbit anti-mouse IgG1, HRP (ThermoFisher Scientific PA1-86329); chicken anti-human actin (beta) antibody (USBiological A0760-43);

goat anti-Mouse IgG, HRP (ThermoFisher Scientific A10551); donkey anti-rabbit gold-conjugated (18 nm) antibody (Jackson ImmunoResearch Laboratories 711-215-152); goat anti-mouse IgG gold-conjugated (15 nm) (Ted Pella #15752-EM.GMHL15); rabbit anti-chicken IgG gold-conjugated (10 nm) (Ted Pella #15971-EM.RCHL10); Goat anti-rabbit IgG (H + L) cross-adsorbed secondary antibody, Alexa Fluor 488 (ThermoFisher Scientific# A-11008); Goat anti-mouse IgG (H + L) cross-absorbed secondary antibody, Alexa Fluor 488 (ThermoFisher Scientific# A-11001); Goat anti-mouse IgG (H + L) cross-absorbed secondary antibody, Alexa Fluor 532 (ThermoFisher Scientific# A-11002).

**Proteins.** Human Dyn1, rat Dyn2, human Dyn1ΔPRD, and rat Dyn2ΔPRD were expressed using Bac-to-Bac baculovirus expression systems (ThermoFisher Scientific) in Sf21 insect cells[10]. Recombinant human gelsolin (Gsn) (gift from Fumihiko Nakamura, Brigham and Women's Hospital); purified non-labeled G-actin, pyrene labeled G-actin (Cytoskeleton); human uPAR protein (R&D Systems); Rhodamine phalloidin (ThermoFisher Scientific).

**Drugs, small molecules, and other reagents**. Bis-T-23 (Ryngo 1-23), Dynole 34-2 (Abcam); Latrunculin A (LatA), cytochalasin D (CytoD), jasplakinolide (Jasp), rhodamine-phalloidin, rhodamine-transferrin, rhodamine-albumin, Hoechst 33342, ProLong™ Gold antifade reagent (ThermoFisher Scientific). Glutathione Sepharose 4B (GE Healthcare); protein G PLUS-agarose beads (Santa Cruz Biotechnology); TFCH (Bane laboratory, Binghamton University, SUNY)[37]; cisplatin, NG-nitro-L-arginine methyl ester (L-NAME), indomethacin (Sigma-Aldrich); iohexol (Omnipaque, 350 mg iodine/ml, TCI America). For cell-based assays, cisplatin and iohexol were dissolved in media before use; other small molecule solutions were prepared in 100% DMSO and frozen until use.

**Immunocytochemistry for ZO-1 and F-actin**
*Bis-T-23 and cisplatin*. MDCK cells were treated with DMSO (0.1%) or Bis-T-23 (5 μM; 0.1% DMSO) for 1 h before adding cisplatin (35 μM) for 23 h. LLC-PK1 cells were treated similarly with 10 μM cisplatin and Bis-T-23.

*Bis-T-23 and LatA*. MDCK cells were incubated in serum-free media (SFM) for 1 h first. Subsequently, in one set, cells were treated with DMSO (0.1%) or Bis-T-23 (30 μM; 0.1% DMSO) for 10 min before adding LatA (0.2 μM; 0.1% DMSO) for 20 min. In another set, cells were treated with identical concentrations of DMSO or LatA for 10 min before adding Bis-T-23 for 10 min. Cells treated with identical concentrations of only DMSO, Bis-T-23, or LatA were used as controls.

*Bis-T-23 and iohexol*. HK-2 cells were treated with media, DMSO (0.1%), or Bis-T-23 (10 μM; 0.1% DMSO) for 1 h. The media was replaced with only media or media containing identical concentrations of iohexol (250 mg/mL)+DMSO or iohexol +Bis-T-23 for 3 h.

*Dynamin mutants overexpression or knockdown*. Multiple dynamin mutants were adenovirally expressed in MDCK, mIMCD, and mIMCD-Dyn2KD cells[10]. Where indicated, cells were treated with 0.1% DMSO or 30 μM Bis-T-23. At the end of all treatments, cells were rinsed with 10 mM sodium phosphate buffer containing 0.9% NaCl, pH 7.4 (PBS), or Dulbecco's phosphate buffered saline (DPBS, Sigma-Aldrich), fixed with 4% paraformaldehyde (PFA), rinsed with PBS or DPBS again, stained with Rhodamine Phalloidin (1:250) and/or ZO-1 antibody (1:70), and mounted with ProLong™ Gold antifade reagent.

**Endocytosis assay**. Post drug treatment, media was replaced with SFM containing rhodamine transferrin (50 μg/ml) for 10 min. The cells were then processed for imaging.

**Oxidative stress-induced carbonylation assessment using TFCH assay**
*Bis-T-23 and LatA*. MDCK cells were treated with DMSO (0.1%, vehicle for Bis-T-23 or LatA) or Bis-T-23 (10 μM) for 20 min before adding DMSO or LatA (0.2 μM) for 10 min. TFCH (20 μM, 0.5% DMSO) was added for 30 min, and samples were processed for imaging[37].

*Bis-T-23 and cisplatin*. MDCK cells were treated with DMSO (0.1%) or Bis-T-23 (5 μM, 0.1% DMSO) for 1 h before adding media or cisplatin (5 μM) for 22.5 h. TFCH was added for 30 min, and samples were processed for imaging[37].

*Bis-T-23 and iohexol*. HK-2 cells were incubated with only media or media containing DMSO (0.1%) or Bis-T-23 (10 μM, 0.1% DMSO) for 1 h. The media was replaced with only media or media containing identical concentrations of iohexol (100 mg/mL)+DMSO or iohexol+Bis-T-23. After 3 h, the cells were washed with fresh media for 2–3 min, exposed to media containing TFCH, and processed for imaging.

*Bis-T-23, suPAR, and iohexol*. HK-2 cells were incubated with media containing vehicle (0.2% PBS + 0.1% DMSO), suPAR (10 ng/mL)+vehicle or suPAR+Bis-T-23 (10 μM)+vehicle for 21 h. Subsequently, Iohexol (100 mg/mL) was added to a subset of samples for an additional 3 h before washing and performing the TFCH assay.

**MDCK, mIMCD, mIMCD Dyn2KD, HK-2 and LLC-PKI cell imaging, and quantification**. All images were captured using a LSM 5 PASCAL or a 600 LSM (Zeiss). For image quantification, all images within an experimental set were acquired using identical optical parameters, thresholded, and analyzed using Fiji (Image J) or Image J (NIH) software. To measure the continuity of tight junctions, a free line was drawn along ZO-1 staining. The graph represents a sum of the length of ZO-1 staining as a percentage of the total length of the cell membrane. A defined region of interest (ROI) drawn with the rectangle tool on the ZO-1-associated cell membrane was quantified to assess ZO-1 intensity. For MDCK cells, the intensity of the F-actin within each cell was quantified. For mIMCD or mIMCD-Dyn2KD cells, the intensity of F-actin in a fixed ROI within each cell was quantified. Cellular carbonylation level was determined by quantifying TFCH fluorescence per cell[37]. In the scatter dot plots for TFCH assays, each dot represents the average intensity of 1–14 cells. The representative images showing cellular

carbnonylation levels were processed identically within a set and assigned a pseudocolor for visual clarity. The staining intensity of treated samples was normalized to the non-treated or vehicle control. An unpaired two-tailed *t*-test or one-way ANOVA with Tukey's multiple comparisons test was performed using Prism (GraphPad). *$P < 0.05$, **$P < 0.01$, ***$P < 0.001$, ****$P < 0.0001$; whereas $P > 0.05$ was considered not significant.

**Total internal reflection fluorescence microscopy (TIRFM)**. MDCK cells stably expressing eGFP-tagged clathrin light chain were treated with DMSO (0.1%), Bis-T-23 (30 and 60 μM), and Dynole (50 μM) for 30 min before data acquisition using TIRFM[18,19].

**Actin polymerization assay using cell extracts**. The cytosolic protein extract (100 μg) was treated with dynamin modulators: Bis-T-23 (10 μM), Dynole (10 μM), or DMSO (0.5–1%), for 10 min, or actin modulators: LatA (0.2 μM), CytoD (0.25 μM), or Jasp (1 μM) for 20 m. Actin polymerization was initiated by adding actin polymerization buffer (10 mM Tris-HCl, pH. 7.4, 100 mM KCl, 2 mM MgCl₂, 0.5 mM ATP, 0.2 mM CaCl₂) and pyrene G-actin (20%, 1 μM). Changes in pyrene actin fluorescence were monitored (excitation: 365 nm; emission: 407 nm) using a cytation5 imaging reader (BioTek).

*Immunodepletion of endogenous Dyn2 from MDCK cells*. Cytosolic extracts were incubated with anti-Dyn2 antibodies (1 μg/200 μg cytosolic proteins) at 4 °C for 2 h, followed by incubation with protein G PLUS-agarose beads at 4 °C for 2 h. After centrifugation, the supernatant was used as the ΔDyn cytosol for the actin polymerization assay. Mock-depletion was performed in the same manner with a non-specific mouse anti-rabbit IgG antibody (1 μg/200 μg cytosolic proteins). The efficacy of the depletion was confirmed to be >90% by Western blot using the same antibody.

**Cell polarity assessment using a transwell system**. MDCK cells were grown in collagen-coated transwells (Corning) for 48 h before treatment with DMSO (0.1%) or Bis-T-23 (30 μM; 0.1% DMSO) for 1 h and cisplatin (50 μM) or media for another 23 h. A Millicell ERS Voltohmmeter was used to measure the transepithelial electrical resistance (before and after 24 h of initiating the treatment) in each sample.

**Atomic force microscopy (AFM)**. MDCK cells were seeded on 34 mm culture dishes (TPP) and grown overnight to confluency. Cells were treated with DMSO (0.1%), LatA (0.2 μM), CytoD (0.25 μM), Bis-T-23 (30 μM), or Dynole (10 μM) for 30 min before data acquisition. When indicated, cells were treated with identical concentrations of either DMSO or Bis-T-23 for 10 min before adding LatA for 20 min. When indicated, experiments were performed in cells expressing either Dyn1WT or Dyn2KE.

*NanoWizard IV system*. AFM imaging was performed using a NanoWizard IV (JPK Instruments) mounted on an Axiovert 200 inverted light microscope (Carl Zeiss). Microscopes were equipped with a heated stage (37 °C) and cells were imaged in HEPES-buffered medium. As previously reported for MDCK cells, indentation measurements were performed with silicon nitride cantilevers (MLCT, Bruker) with a nominal force constant of 0.01 N/m[21]. Automated calibration was performed in contact-free mode as provided by the calibration manager in the JPK SMP software before every series of experiments. Areas of 2500 μm² were scanned in QI imaging mode to acquire 3600 independent force measurements per area. Cells were indented up to a force of 0.9 nN at a vertical speed of 18 μm/s. Determination of Young's Modulus was performed using JPK data analysis software. Fit parameters were set to the Hertzian model according to Sneddon. Cantilever was specified as quadratic pyramid tip shape with a half-angle to edge of 15°. Image analysis and quantification of apical stiffness and stiffness at cell–cell contacts were performed using Fiji software. Topography maps of confluent MDCK cells show elevation at cell–cell junctions. Cell contact sites were marked as ROI manually in the grayscale topography map and ROIs were transferred to the associated force map to analyze stiffness at the respective sites. For determination of apical cell stiffness, grayscale topography maps were analyzed using Fiji threshold presets MaxEntropy/Dark background to mark cells and identify ROIs. ROIs were transferred to the associated force map to determine cell stiffness. Data from at least 10 cells in at least three culture dishes were included in the analysis. Statistical analysis was performed as univariate analysis, defining a *p*-value of <0.05 as statistically significant.

*Bioscope II AFM system*. AFM imaging was performed using a Bioscope II AFM (Veeco) mounted on an Olympus IX 73 inverted light microscope (Olympus). The microscope was equipped with a heated stage (37 °C) and cells were imaged in HEPES-buffered medium. Measurements were performed in contact mode using silicon nitride cantilevers with a nominal force constant of 0.01 N/m. Cantilevers were calibrated before every series of experiments by thermal tune using the manufacturer's software. Force indentation curves were analyzed according to the model of Discher et al.[23] computed with MATLab software. Data from at least 10

cells in at least three culture dishes were included in the analysis. Statistical analysis was performed as univariate analysis, defining a p-value of <0.05 as statistically significant.

**Electron microscopy (EM)**. All EM images were acquired at the Massachusetts General Hospital (MGH), Harvard Medical School, or Brandeis University EM core facility.

*Scanning electron microscopy (SEM)*. MDCK cells were treated as described in the "Immunocytochemistry for ZO-1 and F-actin" section, fixed for 2 h in 2% glutaraldehyde in 0.1 M sodium cacodylate buffer, pH 7.4 (Electron Microscopy Sciences, EMS), rinsed with 0.1 M sodium cacodylate buffer, treated with 1% osmium tetroxide, rinsed in 0.1 M sodium cacodylate buffer and dehydrated through a graded series of ethanol reaching 100%. The samples were dehydrated in the critical point dryer (CPD) Autosamdri®-815 (Tousimis), coated with a 5 nm layer of platinum using a sputter coater (SC) EM ACE600 (Leica), and mounted on stubs using double-sided carbon conductive tape (EMS). Samples were examined at 3–5 kV using S-4700 FE-SEM (Hitachi). Cell height and microvilli length were measured using SEM software and ImageJ, respectively. Microvilli density in a constant area ($0.7 \text{ in}^2$) on the membrane (≥10 areas per condition) was counted manually.

*Transmission electron microscopy (TEM) experiments*. (1) Negative staining of proteins in the reconstituted system: G-actin, F-actin and gelsolin-actin filaments (gelsolin:actin = 1:50) were prepared as described previously[10]. Gelsolin-actin was diluted to 10 μM and incubated with 2 μM recombinant Dyn1, Dyn2, or Dyn1ΔPRD for 60 min at 25 °C in the presence or absence of Bis-T-23 (0.4 μM). All samples were diluted to 0.2 μM final actin concentration, and absorbed to glow discharged formvar-carbon coated copper grids for 1 min, blotted to remove excess solution, negatively stained with 1.5% (w/v) uranyl acetate for 1 min, blotted again, and allowed to air-dry. Images were captured at an acceleration voltage of 80 kV using an FEI Morgani 268 or JEOL JEM 1011 transmission electron microscope (TEM) equipped with CCD camera. The brush tool with 50% opacity in Photoshop (Adobe), and "contour plotter" in ImageJ were used for data presentation.

(2) Platinum replica EM and immunogold EM in cells: MDCK cells were treated as described in the "Immunocytochemistry for ZO-1 and F-actin" section, detergent extracted and processed for PR-EM as described previously[71]. For dynamin localization, samples were fixed with 0.2% glutaraldehyde in 0.1 M sodium cacodylate buffer, pH 7.4, quenched with NaBH₄ for 10 min, treated with Dyn2 antibody (1:100), and subsequently with 18 nm colloidal gold-secondary antibody (1:5). Cells were fixed and processed the same for PREM[71]. For PR-EM and immunogold PREM, samples were dehydrated in a CPD, coated with a 2 nm layer of platinum, and stabilized with 5 nm of carbon using SC. Samples were examined as described above and images were presented as inverted contrast.

(3) Immunogold of ultrathin sections in cells: The experiments were performed as described[8]. Hudy 1 (1:100) and goat anti-mouse IgG gold-conjugated (15 nm) antibodies (1:15) were used for identifying dynamin; and anti-actin (1:200) and rabbit anti-chicken IgG gold-conjugated (10 nm) antibodies (1:15) were used for identifying actin. Sections were examined at 80 kV in a JEOL JEM 1011 TEM with AMT digital camera.

**Measurement of cellular bioenergetics**. A Seahorse Extracellular Flux (XFe24) Analyzer (Agilent) was used to measure oxygen consumption rates (OCR) in real time[47]. HK-2 cells were treated with only media (control) or media containing human recombinant uPAR (10 ng/ml), human anti-uPAR antibody (50 ng/ml), or Bis-T-23 (10 μM) either alone or in combination for 24 h. Respiratory parameters were quantified, using the Cell Mito Stress Test Kit (Agilent), by subtracting respiration rates at times before and after the addition of mitochondrial activators and inhibitors[47]. Experiments were replicated in five wells, averaged for each treatment group and the data were presented as means ± standard errors of the means (SEM). One-way ANOVA using Prism was used to compare controls (CTL) with treatment groups (*P < 0.05, **P < 0.01, ***P < 0.001).

**Rat kidney tissue slice experiment**. Male Spraque-Dawley rats (strain code 400) were from Charles River. Rat kidney were prepared with minor modifications[29]. The slices were treated with DMSO (0.1%) or Bis-T-23 (30 μM, 0.1% DMSO) for 1 h before adding cisplatin (200 μM) for 8 h and fixing with periodate-lysine paraformaldehyde overnight at 4 °C. For the albumin uptake assay, kidney slices were incubated in the presence or absence of Bis-T-23 (30 μM) for 1.5 h before adding rhodamine-albumin (100 μg/ml) for 30 min. Slices were then rinsed with PBS, fixed, and processed. Animal experiments were approved by the institutional committee on Research Animal Care at MGH, in accordance with the NIH Guide for the care and use of the laboratory animals (protocol #2012N000004 (PI, Brown)).

**Murine models**

*1. Single-dose cisplatin induced kidney injury[39]*. Male and female C57BL/J6 mice (9 weeks, 25–30 g) (Strain #000664, common name BL6, The Jackson Laboratory) were randomly divided in 3 experimental groups as follows: (1) cisplatin +DMSO; (2) cisplatin+saline; and (3) cisplatin+Bis-T-23. Mice in each group received a single intraperitoneal (i.p.) injection of 15 mg/kg cisplatin in 0.9% saline. Bis-T-23 dissolved in 1% DMSO was injected i.p. once 24 h before cisplatin injection, and daily at 20 mg/kg per injection. Body weight was monitored, and urine and blood parameters were assessed at baseline and after cisplatin injection. Mice were euthanized 4 days after cisplatin injection by cervical dislocation. Experiments were performed at Rush University and were approved by Rush University Institutional Animal Care and Use Committee (IACUC) (protocol #18-060 (PI, Reiser)).

*2. Iohexol-induced acute kidney injury[47]*. BL6 mice (males and females) were purchased and mice expressing suPAR from the fat tissue (suPAR-Tg)[48] were bred at Rush University. Mice aged 9–12 weeks (25–30 g) were randomly divided in 3 experimental groups as follows: (1) contrast+DMSO; (2) contrast+saline; and (3) contrast+Bis-T-23. Blood and urine were collected for baseline measurements. Mice were denied free oral water access for 48 h. Then, group 1 was administered DMSO (1%, 200 μl), group 2 received 0.9% saline (200 μl) and group 3 was given Bis-T-23 (20 mg/kg in 200 μl of 0.9% saline). After 5 h, all the three groups received L-NAME (10 mg/kg in 100 μl of water), a nitric oxide synthase inhibitor, and indomethacin (10 mg/kg in 100 μl of water), an inhibitor of prostaglandin synthesis. After 1 h, all three groups received iohexol (5 g/kg of water), a non-ionic low osmotic contrast medium. Subsequently, oral rehydration was resumed. A second dose of Bis-T-23 (20 mg/kg in 200 μl of 0.9% saline) was administered to group 3 after 24 h of the first dose. Group 1 and group 2 received DMSO or saline as described above. All injections were given i.p. Urine and blood samples were collected 24 and 48 h after iohexol injection. Experiments were performed at Rush University and were approved by RU IACUC (protocol #19-014 (PI, Reiser)).

*3. Alport syndrome mice*. Alport strain (#002908 Jackson Labs) heterozygous mice were bred at the Rush University, and the homozygous mice ($Col4a3^{-/-}$)[55] were treated with either DMSO (1%) or Bis-T-23 (20 mg/kg) daily starting at 42 days of age (D0 of the treatment). Experiments were performed at Rush University and were approved by Rush University IACUC (protocol #18-060 (PI, Reiser)).

*Survival curves*. Log-Rank (Mantel-cox) test was performed in Prism to analyze survival curves of the two treatment groups and the control animals. The log-rank test reports chi-square value, to compute P-value testing of the null hypothesis that there is no linear trend between column order and median survival.

*Assessment of renal function*. BUN levels were measured using Quantichrom Urea assay kit (BioAssay Systems). Urinary albumin-to-creatinine ratio (ACR) was assessed by mouse albumin ELISA kit (Bethyl Laboratories) and creatinine colorimetric assay kit (Cayman Chemical). Assays were performed according to manufacturer's protocols and read using EnSpire multimode plate reader (Perkin Elmer). Unpaired two-tailed t-tests were performed using Prism (*P < 0.05, **P < 0.01, ***P < 0.001, ****P < 0.0001; whereas P > 0.05 was considered not significant, ns).

## Data availability

The authors declare that all data supporting the findings of this study are available within the paper, its supplementary information files, and in the Source data file. There are no restrictions on any data availability. We did not use any accession codes, unique identifiers, or web links for publicly available datasets. No clinical datasets were used.

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

## Acknowledgements

We would like to thank Astrid Schauss and Lisa Wirtz (CECAD Imaging Facility, University of Cologne, Germany) for their advice on AFM measurements and analysis. We would like to thank Benjamin Dellaripa for his technical assistance with cell culture and image quantification. We acknowledge support from the National Institutes of Health (R01 DK093773 and DK087985 to S.S.; R01 DK084195 to V.G. and S.S.; GM73165 to M.M.; CA227747 to S.L.B.). C.G. was supported by Rush University Medical Center. Electron microscopy was performed in the Microscopy Core of the Center for Systems Biology/Program in Membrane Biology, which is partially supported by an Inflammatory Bowel Disease Grant DK043351. P.T.B. and H.H. received funding from the DFG (KFO329, BR-BR 2955/8-1), the Koeln Fortune Program of the University of Cologne.

## Author contributions

K.M. and S.S. conceptualized the project, wrote the paper, and assembled the figures. K.M. carried out cell-based assays for IF, EM, and biomolecule carbonylation analysis. A.C. carried out EM experiments. C.G. carried out IF and actin polymerization experiments. Y.S. performed the mice experiments as well as Scr and BUN analyses. C.G. and R.B. carried out experiments using kidney slices. B.P. carried out IF and TER experiments. M.M. carried out the TIRFM experiments. P.B. and H.H. carried out AFM experiments. B.S., X.W., and Y.R.S. carried out AKI experiments in mice. M.M.A. carried out Seahorse extracellular flux experiments. S.L.B. provided TFCH and knowledge concerning OS-induced carbonylation. V.G. and J.R. provided knowledge with regard to kidney epithelial cells and D.B. provided knowledge regarding AKI. All authors discussed the results and the manuscript.

## Competing interests

S.S. and J.R. are co-founders and shareholders of Walden Biosciences, a biotechnology company that develops novel kidney-protective therapies. S.L.B. and K.M. are inventors on a pending patent application pertaining to the detection of oxidative stress-induced carbonylation. The remaining authors declare no competing interests.
