## [Peer Review File · Nature Communications]

REVIEWER COMMENTS

Reviewer #1 (Remarks to the Author):

Using an MDCK cell model, the authors propose that the GTPase dynamin crosslinks actin filaments into branched networks within the cortical actin cytoskeleton, which in turn increases cortical tension at the apical cell surface and contributes to the establishment of renal epithelial cell polarity. These conclusions are partly based on AFM indentation measurements, which show that inhibiting actin polymerization by latrunculin A (LatA) or Cytochalasin D (CytD) reduces cortical tension, while Bis-T-23, an activator of dynamin oligomerization, significantly increases cortical tension.

In my review I was asked by the editors to comment specifically on the AFM measurements conducted in this study, and while having read through the entire manuscript I will therefore restrict my comments to this aspect.

In principle AFM indentations measurements, usually either performed with a conical or microbead indenter, are a useful technique for analyzing cell mechanics. Quantitative QI mode imaging, which essentially performs a series of such indentation measurements in an automated or semi-automated manner over a pre-set grid area using a nanoscale indenter, has become a valuable tool to visualize local variations in cell surface mechanics within reasonable time scales. As such, the employed experimental approach could be suitable to support the authors' claims.

However, in my opinion in the presented study there are severe uncertainties related to the exact experimental procedures, data analysis, and the conclusions drawn that would need substantial revision and additional experiments data to be convincing.

Main points

Measuring cortical cell tension by QI mode

AFM indentation measurements primarily test cell stiffness. From the obtained indentation force curves, useful mechanical parameters, such the Young's (or E modulus) can be extracted by applying a Hertz fit (for spherical indenters) or different modified Hertz fits (Sneddon etc.) for conical indenters. The vast majority of AFM studies report E modules values, either averaged over an entire cell when using cell-sized microbead indenters, or locally resolved values when using QI mode. In the submitted study, the authors however present "cortical tension" and "membrane tension" maps. To me this is an unusual representation and, if available, the authors should cite relevant previous literature using a similar approach to support their analysis. Presenting the data as "cortical tension" instead of the commonly used elastic modulus in my view poses a number of complications. Cortical tension undoubtedly contributes to the measured stiffness values, but there are also other potential contributing factors, such as the vertical compression of possibly untensed submembraneous structures, or even elements of the actin cytoskeleton (not all parts of the actin cytoskeleton are subject to actomyosin contractility). Thus, it is not immediately clear how the authors derive cortical stiffness values from the obtained indentation force curves. Furthermore, the authors neglect to distinguish between the potential contributions of "membrane tension" and "cortical tension". Here it is especially confusing that they use both terms interchangeably (see Fig 1c cortical tension maps and membrane tension plots).

Most studies measuring cell cortex tension have either used micropipette aspiration, AFM "wedge" compression, or microbead indentation measurements. The majority of these studies use a rather large tension probe, assume isotropic tension, and extract a global cell-cortex tension value for the entire cell. For instance, Krieg et al. NCB 2008 extracted single-cell cortex values from force-indentation curves using a cortical shell-liquid core or liquid droplet model. In contrast, the authors in this study present locally resolved tension maps (15x15 μm^2 , 8000 pixel), which would assume that tension is not isometric at least on the submicrometer scale. In principle, such a novel finding would be interesting, but it would require considerably more

extensive methods information to validate this approach. The cited Discher and colleagues paper on which the analysis is based appears to be an exception to many other studies in that it also uses a nanoindenter (AFM tip). However, again that studies explicitly assumes isotropic membrane tension in their cell system, and thus appears to be unsuitable to produce locally resolved tension maps. It is therefore unclear if this model can be applied here.

To back up their analysis the authors should therefore provide a detailed description (1) of their QI parameters (indentation force, force curve length, acquisition frequency/tip speed etc.) and (2) provide a detailed description of their force curve analysis approach to obtain the tension maps and average cell values.

Presented AFM tension maps

The presented surface maps are somewhat inconclusive, since it is unclear where exactly on the cell surface they were obtained. Together with the tension maps, the authors should provide the corresponding topographies to enable the reader to understand the location of the maps. Ideally, optical images of the cells should be provided as well, since it is interesting to know if the measured MDCK cells were part of a confluent monolayer (and this polarized), or single unpolarized cells. In addition, E modulus maps could also be informative.

It is also unclear how the tension values were extracted for "intercellular contacts" and other apical membrane regions. A clear way of classification should be provided. It is also unclear how the cell averaging was performed (were intercellular contacts and apical membrane areas included?).

Experimental detail

The manuscript and the supplementary "Materials and Methods" document are missing important experimental detail (see above) and are partly very confusing. For instance there is mentioning of experiments performed with the Bioscope II instrument in "contact mode" (I did not see any AFM contact mode images), while the "force indentation" curves were analyzed according to Discher and colleagues. Please clarify if this microscope was used for indentation measurements, and if the Discher model was applied to standard indentation measurements using the MLCT cantilevers, or to the QI data recorded with the JPK Nanoscope 4. Which data originates from the JPK software analysis?

Additional points:

I believe reference 23 is wrong – it does not refer to an AFM-relevant paper.

Line 136: The authors state the MDCK exhibit "high levels of cortical tension at the apical membrane and at the intracellular contacts". The authors should provide references to back up this claim, i.e. compare the tension values they determine to other cell systems. In general, a short discussion of their tension values (about 1-5 nN/ μm) with previously determined tension values in other cell systems would be helpful.

Line 136: "A dramatic decrease in cortical tension on both membranes". I assume the authors refer to the apical membrane and sites of cellular contacts, but the wording is somewhat unclear and should be revised.

Line 138: "In contrast, Bis-t-23 significantly increased cortical tension, consistent with the observed positive effect on the density of actin networks and increase in cell height". Could the authors provide relevant references to studies demonstrating a link between cortical actin density and cell height to increased cortical tension? This would further back up their argument.

Line 143: what is the explanation for the two cell population (one with further decreased and one with significantly increased cortical tension) after Bis-23-T followed by CytoD treatment?

Reviewer #2 (Remarks to the Author):

In this study, the authors report the evidence that GTPase dynamin crosslinks actin filaments into branched networks that underlie the cortical actin cytoskeleton and cortical tension at the apical membrane. Using cell and animal models, they further show Bis-T-23, an allosteric activator of actin-dependent dynamin oligomerization, can protect against acute injury. In addition, Bis-T-23 reduced podocyte and tubular injury in the murine model of Alport syndrome (AS).

Overall, this is an interesting study with important observations. The finding of dynamin crosslinking of actin filaments is an advance in cell biology, and the finding of the protective effect of Bis-T-23 has therapeutic implications for both acute and chronic kidney diseases.

There are a couple of issues that can be addressed to further improve the study for publication.

- 1. In the animal disease models, it would be nice to verify the actin filament changes and the effect of Bis-T-23.**
- 2. In the Iohexol-induced AKI study, the protective effect of Bis-T-23 was linked to mitochondria, but it is unclear how it would work on mitochondria, directly or indirectly via actin cytoskeleton?**
- 3. The effect of Bis-T-23 on cisplatin induced AKI is not very impressive. Is the moderate protective effect related to the cisplatin dosage tested?**

Reviewer #3 (Remarks to the Author):

In this manuscript, Mukherjee et al. investigated whether pharmacological activation of the large GTPase dynamin can attenuate kidney renal tubular injury using several murine models. The authors first demonstrated that dynamin is required for establishing renal tubular epithelial cell polarity by crosslinking branched actin network at the apical membrane of the MDCK cells. They also showed that a dynamin agonist, Bis-T-23, can maintain the MDCK cell polarity, not by increasing actin polymerization and independent of endocytosis. Rather, the authors showed that the effect of Bis-T-23 on cell polarity is by promoting crosslinking, which results in increasing the overall network density. Furthermore, Bis-T-23 administration protected kidney function in several injury models.

The strength of the paper lies in the positive effect of Bis-T-23 in protecting kidney function in injury models. However, the mechanism underlying this protection remains unclear. Some of the conclusions are not well supported by the results, especially in the cases where conflicting data are presented.

Major comments:

- 1. The authors concluded that the dynamin crosslinks branched cortical actin networks by direct dynamin-actin interaction, but not by crosslinking actin into bundles. However, how direct dynamin-actin interaction leads to branched actin crosslinking remains unclear. The negative stain EM images in Fig. 1b seems to suggest that dynamin forms rings to link short actin filaments. But it is unclear how the dynamin rings crosslink these filaments. The authors suggested that crosslinking may be mediated by the interaction between actin and the Middle domain of dynamin. But the relatively low resolution of these images does not allow the readers to see how the Middle domain in the dynamin ring interacts with the actin filaments. In contrast, Fig. S4 seems to suggest that dynamin forms helices that bundle actin filaments along the outer rim of the helices, which contradicts the author's claim of Middle domain-actin interaction.**
- 2. The author concluded that dynamin's ability of crosslinking F-actin into bundles is not required for the effect of Bis-T-23 on cell polarity. This conclusion is based on the result that**

overexpression of Dyn Δ PRD in wild-type cells did not alter cell polarity. However, it is likely that the endogenous dynamin would function to maintain cell polarity in these cells, as long as the overexpressed Dyn Δ PRD does not interfere with the function of the endogenous dynamin. If PRD is involved in actin bundling, it is likely that Dyn Δ PRD would not interfere with the actin bundling activity of the endogenous dynamin. The authors should perform the Dyn Δ PRD overexpression experiment in dynamin knockout cells, or at least knockdown cells, to see whether Dyn Δ PRD can rescue the polarity defect in these cells. Short of this, it is premature to conclude that PRD-actin interactions are not essential for renal cell polarity.

3. It is unclear why Bis-T-23 has distinct effects on actin polymerization in podocyte cell extracts and MDCK cell extracts (Fig. S1b). How to interpret this data? Is there any difference in dynamin expression level in podocyte vs. renal tubular epithelial cells? Are different dynamins (1, 2 and 3) expressed in these two cell types?

4. The authors stated that Bis-T-23 is known to influence interactions between dynamin's Middle domain and F-actin. How does it do it? Does it also influence PRD-actin interactions?

5. The authors concluded that Bis-T-23 protects renal tubular cells by enhancing the crosslinking of actin networks. Due to the presence of actin bundles in microvilli at the apical membrane, the effects of Bis-T-23 on dynamin-mediated actin bundling in microvilli should also be investigated to reveal the mechanism by which Bis-T-23 protects kidney cells from AKI.

6. The authors show that dynole, a dynamin inhibitor targeting the GTPase domain and inhibiting GTPase activity, has no effects on F-actin level and ZO1 staining (Fig. S1a). Does this mean that the GTPase activity of dynamin does not play a role in regulating dynamin-actin interaction and thus renal tubular epithelial cell polarity?

7. As the authors mentioned, Cisplatin can induce cellular ATP reduction leading to inhibition of actin polymerization. The authors showed that Bis-T-23 can preserve the cortical actin network from cisplatin-induced injury. Does Bis-T-23-induced dynamin oligomerization resist Cisplatin-induced inhibition of actin polymerization? And how?

8. The authors showed that Bis-T-23 could ameliorate suPAR-enhanced Iohexol-induced AKI. How does actin cytoskeleton contribute to this process?

Minor comments:

1. In Fig. S1a, treatment with cytochalasin D caused a more severe decrease of F-actin level than treatment with latrunculin A. However, they showed opposite effects on ZO1 staining – CytoD treatment has no effect on ZO1 staining. How to interpret this discrepancy?

2. In Fig. 1a PR-TEM images, the authors concluded that Bis-T-23 could increase the density of the actin networks. It is not easy to see the changes in density in these images. Quantification is required.

3. In Fig. 1d, addition of Bis-T-23 followed by LatA resulted in significant higher membrane tension than Bis-T-23 alone. How to interpret this data?

4. In Fig. S5, the authors performed immunoEM to determine the localization of Dyn2 on the cortical actin networks. Most of gold particles appear to be associated with residual membranes. A negative control is required to show the specificity of Dyn2 staining.

RESPONSES TO THE REVIEWERS' COMMENTS

We would like to thank the Reviewers for their constructive comments. We tried to address all their concerns and suggestions, and we hope that the manuscript is ready for publication.

Major new data:

1. As requested by Reviewer 1, we have performed measurements of the cell stiffness in MDCK cells using the Nanowizard system. For generating force curves, we applied the JPK software in QI imaging mode. Analysis of force curves was done employing a Hertzian contact model according to Sneddon implemented in the JPK Data Processing Software. Images in **new Fig. 1c** show Young's Modulus maps of a scan area of 50 x 50 μm over a confluent monolayer of cells (**new Fig. S2a**). Details of the technical adjustments for acquisition and analysis of AFM images are discussed point-by-point below. We feel confident that we have addressed all technical concerns by Reviewer 1. In addition, we have moved original data (old Figure 1) generated using BioScope II system to Supplemental Figure 2b-e.
2. As requested by Reviewers 2 and 3, we show that Bis-T-23 was reducing iohexol-induced oxidative stress by stabilizing actin cytoskeleton in HK-2 cells (**new Fig. 5a and S9d**).
3. As requested by Reviewer 3, we show that expression of Dyn2 Δ PRD rescued loss of F-actin in cells in which endogenous Dyn2 was downregulated (**new Fig. S6**). The rescue was at the level observed for wild type dynamin, suggesting that PRD is not essential for the formation of F-actin in polarized epithelial cells.
4. As requested by Reviewer 3, we have quantified the density of the actin networks in MDCK cells visualized by PR-TEM (**new Fig. 4a-c**).
5. As requested by Reviewer 3, we provided a negative control image to demonstrate the specificity of anti-dynamin antibody conjugated with gold (**new Fig. S3a**).
6. The text has been significantly modified in order to address the Reviewer's comments and to comply with the Nature Communication format.

Point-by-point responses to the Reviewers

Reviewer #1:

Using an MDCK cell model, the authors propose that the GTPase dynamin crosslinks actin filaments into branched networks within the cortical actin cytoskeleton, which in turn increases cortical tension at the apical cell surface and contributes to the establishment of renal epithelial cell polarity. These conclusions are partly based on AFM indentation measurements, which show that inhibiting actin polymerization by latrunculin A (LatA) or Cytochalasin D (CytD) reduces cortical tension, while Bis-T-23, an activator of dynamin oligomerization, significantly increases cortical tension.

In my review I was asked by the editors to comment specifically on the AFM measurements conducted in this study, and while having read through the entire manuscript I will therefore restrict my comments to this aspect.

In principle AFM indentations measurements, usually either performed with a conical or microbead indenter, are a useful technique for analyzing cell mechanics. Quantitative QI mode imaging, which essentially performs a series of such indentation measurements in an automated or semi-automated manner over a pre-set grid area using a nanoscale indenter, has become a valuable tool to visualize local variations in cell surface mechanics within reasonable time scales. As such, the employed experimental approach could be suitable to support the authors' claims.

However, in my opinion in the presented study there are severe uncertainties related to the exact experimental procedures, data analysis, and the conclusions drawn that would need substantial revision and additional experiments data to be convincing.

Main points

Measuring cortical cell tension by QI mode

AFM indentation measurements primarily test cell stiffness. From the obtained indentation force curves, useful mechanical parameters, such the Young's (or E modulus) can be extracted by applying a Hertz fit (for spherical indenters) or different modified Hertz fits (Sneddon etc.) for conical indenters. The vast majority of AFM studies report E modules values, either averaged over an entire cell when using cell-sized microbead indenters, or

locally resolved values when using QI mode. In the submitted study, the authors however present “cortical tension” and “membrane tension” maps. To me this is an unusual representation and, if available, the authors should cite relevant previous literature using a similar approach to support their analysis. Presenting the data as “cortical tension” instead of the commonly used elastic modulus in my view poses a number of complications. Cortical tension undoubtedly contributes to the measured stiffness values, but there are also other potential contributing factors, such as the vertical compression of possibly untensed submembraneous structures, or even elements of the actin cytoskeleton (not all parts of the actin cytoskeleton are subject to actomyosin contractility). Thus, it is not immediately clear how the authors derive cortical stiffness values from the obtained indentation force curves. Furthermore, the authors neglect to distinguish between the potential contributions of “membrane tension” and “cortical tension”. Here it is especially confusing that they use both terms interchangeably (see Fig 1c cortical tension maps and membrane tension plots).

Most studies measuring cell cortex tension have either used micropipette aspiration, AFM “wedge” compression, or microbead indentation measurements. The majority of these studies use a rather large tension probe, assume isotropic tension, and extract a global cell-cortex tension value for the entire cell. For instance, Krieg et al. NCB 2008 extracted single-cell cortex values from force-indentation curves using a cortical shell-liquid core or liquid droplet model. In contrast, the authors in this study present locally resolved tension maps (15x15 μm^2 , 8000 pixel), which would assume that tension is not isometric at least on the submicrometer scale. In principle, such a novel finding would be interesting, but it would require considerably more extensive methods information to validate this approach. The cited Discher and colleagues paper on which the analysis is based appears to be an exception to many other studies in that it also uses a nanoindenter (AFM tip). However, again that studies explicitly assume isotropic membrane tension in their cell system, and thus appears to be unsuitable to produce locally resolved tension maps. It is therefore unclear if this model can be applied here.

To back up their analysis the authors should therefore provide a detailed description (1) of their QI parameters (indentation force, force curve length, acquisition frequency/tip speed etc.) and (2) provide a detailed description of their force curve analysis approach to obtain the tension maps and average cell values.

Response: We thank the reviewer for these detailed comments. We agree that the terminology used in the initial manuscript was imprecise and confusing. In all experiments (prior and new) cell stiffness was assessed by AFM indentation measurements. The original experiments were performed using BioScope II device (Veeco) and indentation curves were processed and analyzed using the MatLab Software according to the Discher model (Sen et al. 2005, Biophysical Journal; doi: 10.1529/biophysj.105.063826). Stiffness data are therefore presented as pN/ μm (the original Figs. 1c, 1d and Tables S2 and S3 are current Figs. S2b-e).

We agree that there are other models to determine cell stiffness or elasticity from force distance curves. The Young’s Modulus according to Sneddon is currently a widely used approach. We therefore repeated all relevant conditions in confluent MDCK cells employing NanoWizard4 with the JPK Software in QI imaging mode (see modified Material and Methods section for details) to determine Young’s Modulus from force curves using the JPK data processing software. Young’s Modulus maps are now presented in **new Fig. 1c** and data generated using this approach are shown in new **Figs. 1d, 1e**.

Analysis with the Sneddon approach yielded similar results to what was previously found when applying the Discher model. We have moved the original data (old Fig. 1d-f) to new Suppl. Figs. 2b-e in the revised manuscript as they demonstrate that both approaches yielded similar trends for identical treatments (interplay between LatA and Bis-T-23). We decided to keep the old data as the old approach was used in additional set of treatments such as CytoD, Dynole (current Fig. S2c,d) and cells expressing DynWT and Dyn2K/E (current Fig.S5b). Those data are complementary to the actin and ZO-1 phenotypes shown in current Fig. S1a and S5a-c, and thus provide additional support to the hypothesis that dynamin influences cell mechanics by altering the actin cytoskeleton via its Middle domain.

To avoid confusion, we use the term “cell stiffness” to describe data from both Young’s Modulus analysis as well as the Discher approach.

Finally, images in the original Fig 1c were generated using NanoWizard 4 system and JPK software. In order to simplify the manuscript and to keep clarity, we removed those data from the current version of the manuscript.

We have included all relevant information on QI parameters and data analysis via the JPK Data Processing software in modified Mat & Methods.

Presented AFM tension maps

The presented surface maps are somewhat inconclusive, since it is unclear where exactly on the cell surface they were obtained. Together with the tension maps, the authors should provide the corresponding topographies to enable the reader to understand the location of the maps. Ideally, optical images of the cells should be provided as well, since it is interesting to know if the measured MDCK cells were part of a confluent monolayer (and this polarized), or single unpolarized cells. In addition, E modulus maps could also be informative.

Response: The original surface maps were not easily comprehensible due to the size of the scan area (15x15 μm). Therefore, we decided to remove these surface maps from the manuscript. Instead, we included Young's Modulus maps of MDCK cells of a scan area of 50x50 μm (**new 1c**). We also included optical overview images, including cantilever position and scan area as well as corresponding topography maps (**new S2a**). These data show that experiments were performed on cells grown to confluency.

It is also unclear how the tension values were extracted for "intercellular contacts" and other apical membrane regions. A clear way of classification should be provided. It is also unclear how the cell averaging was performed (were intercellular contacts and apical membrane areas included?).

Response: We thank the reviewer for this important remark. Image analysis is now extensively described in the modified Mat & Methods.

Experimental detail

The manuscript and the supplementary "Materials and Methods" document are missing important experimental detail (see above) and are partly very confusing. For instance, there is mentioning of experiments performed with the Bioscope II instrument in "contact mode" (I did not see any AFM contact mode images), while the "force indentation" curves were analyzed according to Discher and colleagues. Please clarify if this microscope was used for indentation measurements, and if the Discher model was applied to standard indentation measurements using the MLCT cantilevers, or to the QI data recorded with the JPK NanoWizard 4. Which data originates from the JPK software analysis?

Response: We appreciate this justified criticism. We have strictly separated new data sets generated with the NanoWizard IV device and JPK software (**new Fig. 1c-e, S2a**) from the original data sets generated with the Bioscope microscope and MatLab software (**new Fig. S2b-e**) in the revised manuscript. In the past, we could only use BioScope microscope, and only recently we acquired the NanoWizard IV device. We want to emphasize that both experimental approaches resulted in similar trends, thus providing compelling data that establishes the role of dynamin oligomerization in defining cell stiffness via its influence on the actin cytoskeleton.

Additional points:

I believe reference 23 is wrong – it does not refer to an AFM-relevant paper.

Response: We apologize for this error. We have removed this reference.

Line 136: The authors state the MDCK exhibit "high levels of cortical tension at the apical membrane and at the intracellular contacts". The authors should provide references to back up this claim, i.e. compare the tension values they determine to other cell systems. In general, a short discussion of their tension values (about 1-5 nN/ μm) with previously determined tension values in other cell systems would be helpful.

Response: As we do not think that comparison between cell stiffness values for different cell types contributes to our hypothesis (role of dynamin in modifying cell stiffness), we have removed this sentence.

Line 136: "A dramatic decrease in cortical tension on both membranes". I assume the authors refer to the apical membrane and sites of cellular contacts, but the wording is somewhat unclear and should be revised.

Response: Revised wording: “Treatment with LatA resulted in a significant decrease in cell-cell contact stiffness as well as apical cell stiffness in MDCK cells (Fig. 1c-e).”

Line 138: “In contrast, Bis-t-23 significantly increased cortical tension, consistent with the observed positive effect on the density of actin networks and increase in cell height”. Could the authors provide relevant references to studies demonstrating a link between cortical actin density and cell height to increased cortical tension? This would further back up their argument.

Response: When writing this sentence, we were referring to a study by Shawky et al., 2018 in *Development* (doi: 10.1242/dev.161281) as well as study by Chugh et al., 2017 in *Nature Cell Bio* (doi: 10.1038/ncb3525). Both studies showed that the architecture of actin networks (defined by F-actin cortical density, composition and the length of actin filaments) defines cell surface tension/physical mechanics of a cell. As cortical actin is an essential contributor to polarity of epithelial cells, and as its loss leads to a decrease in cell height (our LatA data), we think that the causation between increased cell stiffness, increased density of the cortical actin, and the increased cell height in the presence of Bis-T-23 is justified.

The current sentence reads: “In contrast, Bis-T-23 significantly increased cell stiffness when compared to DMSO vehicle (Fig. 1c-e), consistent with its positive effects on cell height, microvilli number and the density of actin networks (Table 1 and Fig 1b).” We added Shawky et al. reference into the text.

Line 143: what is the explanation for the two cell population (one with further decreased and one with significantly increased cortical tension) after Bis-23-T followed by CytoD treatment?

Response: Based on the current hypothesis, the length of actin filaments defines their mode of crosslinking (Chugh et al., 2017, *Nat. Cell Bio.* doi: 10.1038/ncb3525). Our study suggests that dynamin’s crosslinking capability is dependent on its oligomerization state and the length of F-actin. To demonstrate this point more clearly, we added data using long, non-capped actin filaments (see **new Fig. 3**). Thus, we interpret these data as a combination of CytoD generated actin filaments and dynamin crosslinking capability: eg in some cells, CytoD might have generated either too short or too long filaments for dynamin to efficiently crosslink them, whereas in some cells the length of the filaments might have been optimal for dynamin’s crosslinking activity. That said, since we could not directly correlate the length of actin filaments to the measured cell stiffness (and thus provide direct evidence for our interpretation), we decided to remove those data from the current version of the manuscript. We did keep data showing that CytoD treatment alone resulted in decreased level of cell stiffness (current Fig. S2b, S2c) as they are complementary to the observed loss of F-actin within the cell body in Fig. S1a, and are complementary to the effect of LatA (another actin-specific small molecule that leads to depolymerization of F-actin via different mechanism).

Reviewer #2:

In this study, the authors report the evidence that GTPase dynamin crosslinks actin filaments into branched networks that underlie the cortical actin cytoskeleton and cortical tension at the apical membrane. Using cell and animal models, they further show Bis-T-23, an allosteric activator of actin-dependent dynamin oligomerization, can protect against acute injury. In addition, Bis-T-23 reduced podocyte and tubular injury in in the murine model of Alport syndrome (AS).

Overall, this is an interesting study with important observations. The finding of dynamin crosslinking of actin filaments is an advance in cell biology, and the finding of the protective effect of Bis-T-23 has therapeutic implications for both acute and chronic kidney diseases.

There are a couple of issues that can be addressed to further improve the study for publication.

1. In the animal disease models, it would be nice to verify the actin filament changes and the effect of Bis-T-23.

Response: We wholeheartedly agree with the Reviewer that it would be powerful to follow F-actin changes in the renal tubules of live animals. Unfortunately, this is technically challenging and as such goes beyond our capability. Recently, Corridon et al., *Scientific reports*, 2021 (doi.org/10.1038/s41598-021-87807-6) used intravital two photon microscopy in order to follow changes in actin brush border upon AKI in live rats. We think that our ex vivo images in current Figure 4g are similar to the published data using live animals. Importantly, our study showed that cisplatin induced ~40% decrease in F-actin intensity in brush border (Fig. 4g), which is in line with observed in vivo changes in fluorescent intensity for F-actin obtained from proximal and distal

tubular segments upon AKI in live animals (Fig. 7 in Corridon et al). We have added this reference in the text: “Similar loss of F-actin at the brush border upon onset of severe ischemia-reperfusion injury was recently observed using intravital imaging {Corridon, 2021 #1509}.”

2. In the Iohexol-induced AKI study, the protective effect of Bis-T-23 was linked to mitochondria, but it is unclear how it would work on mitochondria, directly or indirectly via actin cytoskeleton?

Response: It has been shown that Iohexol directly injures mitochondria leading to increase in ROS production (eg Lei, 2018, Cell Physiol Biochem: DOI: 10.1159/000488827). It has been shown that ROS affects the actin cytoskeleton (eg Balta et al., 2020; DOI: 10.3389/fcell.2020.618261). Finally, it has been reported that decreased actin dynamics cause disruption of the mitochondrial membrane and an increase in ROS (Gourlay et al., 2004: DOI: 10.1083/jcb.200310148).

Together, these studies identify crosstalk between the status of the actin cytoskeleton and the mitochondrial physiology in health and disease. As Bis-T-23 is dynamin-specific, the protective mechanism is expected to be indirect and via stabilization of the actin cytoskeleton.

As requested, in **new Figures 5a and 5b**, we show that Bis-T-23 protected actin cytoskeleton from Iohexol-induced depolymerization. Bis-T-23 also protected against Iohexol-induced oxidative damage (carbonylated macromolecules, TFCH assay). Together, these data provide evidence that Bis-T-23-stabilized actin protected against Iohexol induced injury. We have modified the text to reflect these new data and explain this concept.

3. The effect of Bis-T-23 on cisplatin induced AKI is not very impressive. Is the moderate protective effect related to the cisplatin dosage tested?

Response: DMSO (vehicle control) acts as hydroxyl radical scavenger, thus effectively decreasing cisplatin concentration in the cell (Ozkok and Edelstein, 2014; DOI: 10.1155/2014/967826). That is the reason why Bis-T-23 protection seems less pronounced when compared to DMSO treated animals then control animals (Fig. 4h). Similar protective DMSO phenotype was observed by others (Baliga et al., 1998. Kidney international; DOI: 10.1046/j.1523-1755.1998.00767.x).

Reviewer #3:

In this manuscript, Mukherjee et al. investigated whether pharmacological activation of the large GTPase dynamin can attenuate kidney renal tubular injury using several murine models. The authors first demonstrated that dynamin is required for establishing renal tubular epithelial cell polarity by crosslinking branched actin network at the apical membrane of the MDCK cells. They also showed that a dynamin agonist, Bis-T-23, can maintain the MDCK cell polarity, not by increasing actin polymerization and independent of endocytosis. Rather, the authors showed that the effect of Bis-T-23 on cell polarity is by promoting crosslinking, which results in increasing the overall network density. Furthermore, Bis-T-23 administration protected kidney function in several injury models. The strength of the paper lies in the positive effect of Bis-T-23 in protecting kidney function in injury models. However, the mechanism underlying this protection remains unclear. Some of the conclusions are not well supported by the results, especially in the cases where conflicting data are presented.

Major comments:

1. The authors concluded that the dynamin crosslinks branched cortical actin networks by direct dynamin-actin interaction, but not by crosslinking actin into bundles. However, how direct dynamin-actin interaction leads to branched actin crosslinking remains unclear. The negative stain EM images in Fig. 1b seems to suggest that dynamin forms rings to link short actin filaments. But it is unclear how the dynamin rings crosslink these filaments. The authors suggested that crosslinking may be mediated by the interaction between actin and the Middle domain of dynamin. But the relatively low resolution of these images does not allow the readers to see how the Middle domain in the dynamin ring interacts with the actin filaments. In contrast, Fig. S4 seems to suggest that dynamin forms helices that bundle actin filaments along the outer rim of the helices, which contradicts the author's claim of Middle domain-actin interaction.

Response: We apologize for not clearly explaining the rationale for our studies as well as data. We tried to address major concerns by several additions:

- a) In **new Supplemental Fig. 4f** we summarize current knowledge regarding actin binding domains on dynamin and its mechanism for crosslink actin filaments. We mark two identified binding sites for F-actin: one in the Middle domain and one in the PRD, and cite appropriate references in the text. Current studies suggest that dynamin helices form actin bundles whereas dynamin dimers, tetramers and rings form actin networks.
- b) The role of dynamin's PRD (and thus the formation of actin bundles) is addressed by the use of Dyn^{ΔPRD}. In addition, the role of the Middle domain in the formation of actin networks is addressed by the use of dynamin mutant Dyn^{K/E} (reported in Gu et al., EMBO J, 2010). Dyn^{K/E} has been successfully used by us and others when examining direct dynamin-actin interactions (e.g. Chuang et al, JCB, 2019). Keq references have been added to the text.
- c) Data in Supplemental Fig. 3d demonstrate that Dyn^{ΔPRD} crosslinks actin into networks. Thus, PRD-actin interactions are not essential for network formation. In addition, data in Fig. S5 (showing that expression of Dyn^{ΔPRD} rescues actin phenotypes in cells in which endogenous dynamin was downregulated), strongly suggest that formation of actin bundles by dynamin helices is not essential for MDCK cell polarity.
- d) Although EM does not have resolution needed to identify dynamin domain responsible for binding F-actin, we added **new micrographs in Fig. 2d, 2e** to better define distinct dynamin oligomerization states, their shape and size and the number of F-actin bound by each of those distinct dynamin structures.

2. The author concluded that dynamin's ability of crosslinking F-actin into bundles is not required for the effect of Bis-T-23 on cell polarity. This conclusion is based on the result that overexpression of Dyn^{ΔPRD} in wild-type cells did not alter cell polarity. However, it is likely that the endogenous dynamin would function to maintain cell polarity in these cells, as long as the overexpressed Dyn^{ΔPRD} does not interfere with the function of the endogenous dynamin. If PRD is involved in actin bundling, it is likely that Dyn^{ΔPRD} would not interfere with the actin bundling activity of the endogenous dynamin. The authors should perform the Dyn^{ΔPRD} overexpression experiment in dynamin knockout cells, or at least knockdown cells, to see whether Dyn^{ΔPRD} can rescue the polarity defect in these cells. Short of this, it is premature to conclude that PRD-actin interactions are not essential for renal cell polarity.

Response: As requested, we have expressed Dyn2^{ΔPRD} in cells in which dynamin has been downregulated (**new Supplemental Fig. 6**). As MDCK cells are from dog, and as we could not find commercially available dog specific shRNA to downregulate endogenous dynamin in those cells, we generated a mouse IMCD cell line in which expression of endogenous Dyn2 was stably downregulated using lentivirus-based shRNA. Identical construct (shRNA) was successfully used in transient experiments in mouse podocytes (Gu et al., 2010, EMBO J). **New data in Supplemental Fig. 6** show level of endogenous dynamin was reduced by ~85% in IMCD-Dyn2KD cells. Although viable, IMCD-Dyn2KD cells grow slower. They exhibit major loss of stress fibers within the cell body and formation of rings of cortical actin associated with cell membrane (Fig. S6b, S6c). Although those cells were still partially polarized, the smoothness of ZO1 staining was altered and appeared spotty (Fig. S6b). Expression of either Dyn2^{WT} or Dyn2^{ΔPRD} fully rescued stress fibers formation in those cells (Fig. S6d).

In addition, we want to point out that dynamin exists in equilibrium between dimers and tetramers (Muhlberg et al., 1997, EMBO J; DOI: 10.1093/emboj/16.22.6676). Any dynamin mutant expressed in the cell (usually at the level 10 to 100-fold over endogenous dynamin) forms *hetero tetramers* as well as higher order oligomers that are formed by wild type enzyme and mutants, thus "poisoning" the endogenous enzyme. It has been extensively shown that almost all dynamin mutants generated so far exhibit *dominant effects* on endogenous dynamin when overexpressed in cells (e.g. van der Bliet et al, JCB, 1993, 122:553; Damke et al., 2001, Mol Bio Cell, 12:2578). Thus, overexpression of Dyn2^{ΔPRD} in MDCK cells was expected to overcome wild type enzyme and thus generate distinct phenotype(s). For example, McNiven et al, JCB, 2000, (DOI: 10.1083/jcb.151.1.187) showed that overexpression of Dyn2^{ΔPRD} resulted in "striking increase in the number of actin stress fibers... (in fibroblasts)".

Finally, we want to point out that it has been shown that loss of F-actin (by CytoD treatment) resulted in actin aggregates along the cell membrane. As these actin aggregates also contained ZO-1, they prevented to some degree loss of cell polarity measured by ZO-1 staining (Stevenson and Begg, JCS; 1994; PMID: 8006058). We want to point out that we have observed a similar phenotype in IMCD cells in which endogenous dynamin was downregulated (Supplemental Fig. 6b): formation of actin “rings” along the cell membrane and patchy ZO-1 staining (despite the major loss of F-actin). Both phenotypes were rescued by the expression of Dyn2^{WT} or Dyn2^{ΔPRD} (Supplemental Fig. 6d).

3. It is unclear why Bis-T-23 has distinct effects on actin polymerization in podocyte cell extracts and MDCK cell extracts (Fig. S1b). How to interpret this data? Is there any difference in dynamin expression level in podocyte vs. renal tubular epithelial cells? Are different dynamins (1, 2 and 3) expressed in these two cell types?

Response: This is really a great question: what determines cell-specific effects of dynamin on the actin cytoskeleton. We and others are in the process of trying to answer this question. Based on our current knowledge, ability to induce actin polymerization or to crosslink actin filaments into bundles and/or networks is common for all three isoforms. Our study suggests that the cell-type specificity for dynamin’s role in regulating actin cytoskeleton is due to interplay between the status of the actin filaments (their length) and dynamin oligomerization status (which is influenced by dynamin’s interactions with other cell specific proteins).

The length of actin filaments is regulated in cell type specific manner by multiple actin binding and severing proteins including gelsolin. We have shown that dynamin can promote polymerization of *only* gelsolin capped filaments, thus suggesting its role only in cells in which gelsolin plays a major role in regulating actin polymerization (e.g. podocytes, platelets, fibroblasts). Indeed, our current study suggests that dynamin does not provide major effect on actin polymerization in MDCK cells.

In addition, dynamin oligomerization is regulated by its local concentration (dynamin targeting to distinct sites in the cell), and the interactions with SH3-domain containing proteins some of which are actin binding proteins (e.g. cortactin, amphiphysin). Often, there is an interplay between multiple dynamin/ABPs mechanisms as in the case of dynamin-cortactin-Arp2/3 complex-mediated actin reorganization in growth factor-stimulated cells (Krueger et al., 2003; DOI: 10.1091/mbc.e02-08-0466). As we do not directly address this question in our study, we can only suggest several mechanisms in Discussion.

4. The authors stated that Bis-T-23 is known to influence interactions between dynamin’s Middle domain and F-actin. How does it do it? Does it also influence PRD-actin interactions?

Response: We apologize if this was not stated correctly, but Bis-T-23 is dynamin-specific allosteric activator that promotes dynamin-dynamin interactions by binding dynamin (Gu et al., 2014, Traffic). Bis-T-23 does not influence direct dynamin-actin interactions. The effect that Bis-T-23 has on actin is via its effect on dynamin oligomerization. In new Figure 2d we show that number of F-actin bound by dynamin is dependent on its oligomerization state. As Bis-T-23 promotes dynamin oligomerization, oligomerized dynamin can bind more actin filaments. Since the formation of actin bundles is also dependent on dynamin oligomerization into rings and spirals, Bis-T-23 does promote bundling of actin filaments (new Fig. S4c, S4d), which in turn might explain increase in number of microvilli in Bis-T-23 treated cells (Fig. 1a and Table 1). We tried to explain this better in the modified version of the manuscript.

5. The authors concluded that Bis-T-23 protects renal tubular cells by enhancing the crosslinking of actin networks. Due to the presence of actin bundles in microvilli at the apical membrane, the effects of Bis-T-23 on dynamin-mediated actin bundling in microvilli should also be investigated to reveal the mechanism by which Bis-T-23 protects kidney cells from AKI.

Response: As requested, we have added **new Figure S4e**, that shows that dynamin indeed localizes to actin bundles that underlie filopodia (structural equivalent to microvilli). That said, expression of Dyn2^{ΔPRD} did not result in the loss of MDCK cell polarity, and Dyn2^{ΔPRD} restored wild type level of F-actin and ZO-1 staining pattern in IMCD cells (Supplemental Fig. S5 and S6). Since the formation of bundles is mediated by PRD-actin interactions, these data suggest that the *main* effect of Bis-T-23 is on cortical actin networks and not dynamin-mediated bundles within microvilli. As stated in the text “Since microvilli exhibit exquisite length control defined by the cortical actin at their base¹⁸, these data provide evidence that Bis-T-23 modified the cortical actin at the

apical membrane”.

6. The authors show that dynole, a dynamin inhibitor targeting the GTPase domain and inhibiting GTPase activity, has no effects on F-actin level and ZO1 staining (Fig. S1a). Does this mean that the GTPase activity of dynamin does not play a role in regulating dynamin-actin interaction and thus renal tubular epithelial cell polarity?

Response: We and others have shown that dynamin-actin interactions are independent from its GTPase activity (eg Gu et al., EMBO J, 2010; Zhang et al., Nature Cell Bio, 2020). Whether GTPase activity plays a role in the plasticity of the actin dynamics regulated by dynamin remains to be further investigated.

7. As the authors mentioned, Cisplatin can induce cellular ATP reduction leading to inhibition of actin polymerization (what about depolymerization...). The authors showed that Bis-T-23 can preserve the cortical actin network from cisplatin-induced injury. Does Bis-T-23-induced dynamin oligomerization resist Cisplatin-induced inhibition of actin polymerization? And how?

Response: A combination of cell culture data (Fig. 4a-f) and ex vivo data (Fig. 4g), provide evidence that actin networks formed by oligomerized dynamin are indeed resistant to cisplatin-induced depolymerization. We think that the effect/mechanism is similar to the ability of Bis-T-23 to protect actin networks from LatA-induced depolymerization (Fig. 1a).

8. The authors showed that Bis-T-23 could ameliorate suPAR-enhanced Iohexol-induced AKI. How does actin cytoskeleton contribute to this process?

Response: Please see our responses to Reviewer 2, point 2.

Minor comments:

1. In Fig. S1a, treatment with cytochalasin D caused a more severe decrease of F-actin level than treatment with latrunculin A. However, they showed opposite effects on ZO1 staining – CytoD treatment has no effect on ZO1 staining. How to interpret this discrepancy?

Response: We thank the Reviewer for bringing up this contradiction. We have repeated CytoD experiment and show new images and corresponding quantifications in new Supplemental Fig. 1a. CytoD resulted in significant loss of F-actin as well as cell polarity (discontinues ZO-1 staining pattern), although at a lower extent than LatA.

2. In Fig. 1a PR-TEM images, the authors concluded that Bis-T-23 could increase the density of the actin networks. It is not easy to see the changes in density in these images. Quantification is required.

Response: As requested, we have quantified PR-TEM images and data are shown in new Figure 4a-c.

3. In Fig. 1d, addition of Bis-T-23 followed by LatA resulted in significant higher membrane tension than Bis-T-23 alone. How to interpret this data?

Response: Based on the current hypothesis, the length of actin filaments defines their mode of crosslinking (Chugh et al., 2017, Nat. Cell Bio. doi: 10.1038/ncb3525). We have previously shown that short filaments promote dynamin oligomerization in vitro (Gu et al., EMBO J, 2010: DOI: 10.1038/emboj.2010.249) and in live cells (Gu et al., Traffic, 2014: DOI: 10.1111/tra.12178). This study shows that dynamin's crosslinking capability is dependent on its oligomerization state and the length of F-actin (Fig. 3a-c). Together, these data suggest that shortening of the actin filaments by LatA increases dynamin crosslinking capability by promoting dynamin's oligomerization. **New Fig. 3** provides a molecular mechanism for observed cell phenotypes.

4. In Fig. S5, the authors performed immunoEM to determine the localization of Dyn2 on the cortical actin networks. Most of gold particles appear to be associated with residual membranes. A negative control is required to show the specificity of Dyn2 staining.

Response: As requested, we added an appropriate control in the **new Supplemental Figure 3a** (anti-dynamin Ab that was not gold-labeled). We do want to point out that on images shown in Fig. 2f and S3c (gold particles associated with F-actin), the residual membrane was not present.

REVIEWER COMMENTS

Reviewer #1 (Remarks to the Author):

The revised manuscript largely addresses my previous concerns related to the cell stiffness data representation. In my opinion the manuscript now provides a clearer cell stiffness analysis approach, as well as helpful additional methodological explanations. The authors have provided an extensive additional data set using a more commonly used experimental approach. While the new data largely corroborates their previous findings, they also observed subtle differences. For instance, addition of the DMSO vehicle increases membrane tension (previous manuscript), but reduces cell-cell contact stiffness (revised manuscript). Likewise, mechanical enhancement by Bis-T23 appears now lower than before. It would be helpful if the authors could briefly mention these findings and discuss possible reasons for these subtle yet obvious differences. Pending these minor revisions I have no further objection in recommending this manuscript for publication (in regards to the cell stiffness analysis).

Reviewer #2 (Remarks to the Author):

I feel the authors have done a good job in the revision that has addressed the main issues.

Reviewer #3 (Remarks to the Author):

In the revised manuscript, the authors addressed most of the points that I raised earlier. They have included additional data on the measurements of the cell stiffness, Dyn2 Δ PRD rescue experiment in Dyn2 KD cells, the effect of Bis-T-23 on actin cytoskeleton in HK-2 cells, quantification of the actin networks in PR-TEM. There are still several points that I would like the authors to clarify:

- 1. In the new Supplemental Fig. 4f, the authors summarized that Dyn has two actin binding domains, one in the Middle domain and one in the PRD, and they proposed that helices form actin bundles whereas dynamin dimers, tetramers and rings form actin networks. To support this, the authors defined distinct dynamin oligomerization states crosslinking actin filaments in Fig. 2d, 2e, 2f. How did the authors define dynamin dimers, tetramers and rings, especially in Fig. 2f? Does Dyn2 delta PRD also form similar structures to organize actin networks?**
- 2. It is interesting that the authors showed dynamin localization to actin bundles within filopodia (structural equivalent to microvilli) in the new Figure S4e. Given that dynamin helices form actin bundles whereas dynamin dimers, tetramers and rings form actin networks, the data so far suggest that dynamin exists as helices in filopodia or microvilli, but as dimers, tetramers and rings in cortical region at the apical membrane. How does a cell regulate dynamin oligomerization states in these places?**
- 3. In the revised manuscript, the authors used gold-conjugated anti-Dyn2 antibody to label Dyn2 in cells with PR-TEM. Please update the method.**

RESPONSES TO THE REVIEWER'S COMMENTS

Reviewer #1:

The revised manuscript largely addresses my previous concerns related to the cell stiffness data representation. In my opinion the manuscript now provides a clearer cell stiffness analysis approach, as well as helpful additional methodological explanations. The authors have provided an extensive additional data set using a more commonly used experimental approach. While the new data largely corroborates their previous findings, they also observed subtle differences. For instance, addition of the DMSO vehicle increases membrane tension (previous manuscript) but reduces cell-cell contact stiffness (revised manuscript). Likewise, mechanical enhancement by Bis-T23 appears now lower than before. **It would be helpful if the authors could briefly mention these findings and discuss possible reasons for these subtle yet obvious differences.** Pending these minor revisions I have no further objection in recommending this manuscript for publication (in regards to the cell stiffness analysis).

Response: The Reviewer comments on differences observed for DMSO treatment shown in Fig. 1d (marginal drop in cell-cell contact stiffness) and no phenotype shown in Fig. S2d (with some trend toward increase in cell stiffness). In both instances, Bis-T-23 increased cell stiffness over DMSO treatment, though to a greater extent in Fig. S2d. The Reviewer suggests that we mention and discuss the reason for these observations.

While we acknowledge the differences pointed out by the Reviewer, we do not believe that these subtle differences are mechanistically significant. Therefore, we did not comment on them in the previous version of the manuscript. The key point communicated in both these Figures is that Bis-T-23 protects against LatA-driven injury, which is consistently evident at the apical membrane and at cell-cell contact in both Figures. We think that the slight variation in cell-cell contact stiffness for DMSO treated samples is merely an effect of the inherent diversity observed in cell-based assays. Stronger Bis-T-23 phenotype in S1d may also be a consequence of the same. Meager differences in experimental parameters, such as cell density and/or the duration of cells in culture, can result in these observed variations. Although the data in both these Figures were obtained using different experimental approaches, we do not think that this is the reason for the observed heterogeneity. This notion is supported by the identical trends for DMSO and Bis-T-23 regarding cell stiffness at the apical membrane. Importantly, the concept that a dynamin agonist can partially preserve membrane stiffness when injured by an actin depolymerizer holds true irrespective of the technique or assay conditions used.

As suggested by the Reviewer, we have added a few sentences in the Figure legend of S2d to communicate the difference and a possible rationale for it: "Note, cell-cell contact stiffness observed in cells treated with DMSO or Bis-T-23 is slightly different in Fig 1d and Fig S2d. These differences may be due to variations in experimental parameters, such as cell density and/or the duration of cells in culture. Importantly, identical trends for Bis-T-23-mediated preservation of Lat-A injury are seen at the apical membrane and at cell-cell contact in both Figures regardless of the experimental technique used."

Reviewer #2:

I feel the authors have done a good job in the revision that has addressed the main issues.

Response: We thank the reviewer for finding our original revision acceptable for publication.

Reviewer #3:

In the revised manuscript, the authors addressed most of the points that I raised earlier. They have included additional data on the measurements of the cell stiffness, Dyn2ΔPRD rescue

experiment in Dyn2 KD cells, the effect of Bis-T-23 on actin cytoskeleton in HK-2 cells, quantification of the actin networks in PR-TEM. There are still several points that I would like the authors to clarify:

1. In the new Supplemental Fig. 4f, the authors summarized that Dyn has two actin binding domains, one in the Middle domain and one in the PRD, and they proposed that helices form actin bundles whereas dynamin dimers, tetramers and rings form actin networks. To support this, the authors defined distinct dynamin oligomerization states crosslinking actin filaments in Fig. 2d, 2e, 2f. **How did the authors define dynamin dimers, tetramers and rings, especially in Fig. 2f? Does Dyn2 delta PRD also form similar structures to organize actin networks?**

Response: **With regard to Fig. 2f**, the Reviewer is correct in pointing out that we cannot define dynamin oligomerization status in cells using EM and gold-labeling. We have originally attributed dynamin oligomerization state based on the number of actin filaments associated with gold particles: if a single gold particle was associated with two filaments we assumed (based on the in vitro data) that this was a dimer; if a single gold particle associated with four filaments we assumed that this was a tetramer; and if multiple gold particles were associated with multiple actin filaments, we assumed that they were rings.

While our assumptions might have been correct, we agree with the Reviewer that there is a possibility of over-interpreting the data. Therefore, we have removed text defining distinct dynamin oligomerization states in cells and replaced it with label “gold-labeled endogenous Dyn2”.

With regard to Dyn Δ PRD, it is worth noting that all solved crystal structures of dynamin are of dynamin lacking PRD (Dyn Δ PRD proteins): dimer (Faelber et al., Nature 2011); tetramer (Reubold et al., Nature, 2015); helix (Liu et al., Nature Communications, 2021). Thus, Dyn Δ PRD does form all known dynamin oligomerization states and it is a fully functional enzyme with regard to endocytosis. The role of PRD in the formation of tight actin bundles by dynamin helices was suggested by Zhang et al, Nature Cell bio, 2020. Therefore, our model in Supplemental Figure 4f incorporates current hypothesis with regard to the role of dynamin’s PRD in the bundle formation.

2. It is interesting that the authors showed dynamin localization to actin bundles within filopodia (structural equivalent to microvilli) in the new Figure S4e. Given that dynamin helices form actin bundles whereas dynamin dimers, tetramers and rings form actin networks, the data so far suggest that dynamin exists as helices in filopodia or microvilli, but as dimers, tetramers and rings in cortical region at the apical membrane. **How does a cell regulate dynamin oligomerization states in these places?**

Response: The Reviewer is correct in suggesting that the key aspect of dynamin’s effect on F-actin (networks vs bundles) is dependent on its oligomerization state. We state in Discussion: “Dynamin oligomerization is cooperative and is regulated by its concentration⁵⁶, the length of actin filaments (this study, and ¹³), and SH3-domain-containing proteins^{25,57}. The combination of all these mechanisms ultimately defines temporal, spatial, and cell-type specificity of dynamin’s role in modifying and/or establishing diverse actin structures”. For example, it has been shown that Tks5 regulates dynamin oligomerization in invadosomes formed by myoblasts (Chuang et al., JCB, 2019).

3. In the revised manuscript, the authors **used gold-conjugated anti-Dyn2 antibody to label Dyn2 in cells with PR-TEM. Please update the method.**

Response: We thank the Reviewer for drawing our attention to this. We apologize for the confusion we have caused by mis-labeling our figures and figure legends. We did not use gold-conjugated anti-Dyn antibody. Instead, as it was stated in our Methods, we used primary antibodies followed by gold-conjugated secondary antibodies. We have changed labeling in the S3a, S3b to read unlabeled Dyn2 and gold-labeled Dyn2, and we have changed Figure Legends to reflect this changes (highlighted in red).

REVIEWERS' COMMENTS

Reviewer #1 (Remarks to the Author):

The reviewers addressed my remaining concerns and I have no further objections against recommending this manuscript for publication.

Reviewer #3 (Remarks to the Author):

The authors have addressed my concerns in the revised manuscript. I have no further comments.